# Sparsing Law: Towards Large Language Models with Greater Activation Sparsity

## Abstract

Activation sparsity denotes the existence of substantial weakly-contributed elements within activation outputs that can be eliminated, benefiting many important applications concerned with large language models (LLMs), such as computation acceleration and model interpretability. Although promoting greater activation sparsity within LLMs deserves deep studies, existing works lack comprehensive and quantitative research on the correlation between activation sparsity and potentially influential factors. In this paper, we present a comprehensive study on the quantitative scaling properties and influential factors of the activation sparsity within decoder-only Transformer-based LLMs. Specifically, we propose PPL-$p\%$ sparsity, a precise and performance-aware activation sparsity metric that is applicable to any activation function. Through extensive experiments, we find several important phenomena. Firstly, different activation functions (i.e., ReLU and SiLU) exhibit comparable performance but opposite training-time sparsity trends. The activation ratio (i.e., $1 - \text{sparsity ratio}$) evolves as a convergent increasing power-law and decreasing logspace power-law with the amount of training data for SiLU-activated and ReLU-activated LLMs, respectively. These demonstrate that ReLU is more efficient as the activation function than SiLU and can leverage more training data to improve activation sparsity. Secondly, the activation ratio linearly increases with the width-depth ratio below a certain bottleneck point, indicating the potential advantage of a deeper architecture at a fixed parameter scale. Finally, at similar width-depth ratios, we surprisingly find that the limit value of activation sparsity varies weakly with the parameter scale, i.e., the activation patterns within LLMs are insensitive to the parameter scale. These empirical laws towards LLMs with greater activation sparsity have important implications for making LLMs more efficient and interpretable.

## 1 Introduction

Activation sparsity refers to the phenomenon where considerable elements within the output of a neural layer (typically activation functions, as shown in Figure 1) are zero or low values and thus contribute weakly to the final model output given a specific input. As a prevalent property of many language and vision modeling architectures (Li et al., 2022), activation sparsity has wide practical values, such as inference acceleration (Liu et al., 2023; Song et al., 2023; Xue et al., 2024; Song et al., 2024a), training acceleration (Zhang et al., 2024b), and LLM interpretation (Sajjad et al., 2022; Zhang et al., 2023). Generally, a model with a greater sparsity ratio (i.e., the ratio of inactivated elements) has more potential in these scenarios. However, this raises an underexplored problem: *how to obtain an LLM with greater activation sparsity?*

A simple solution is to design constraints at the model architecture level that force the model to have a predefined large sparsity ratio. For example, mixture-of-experts (MoE), the most popular architecture-constrained design, typically uses a token-level top-k parameter selection router to assign a fixed sparsity ratio for each token at each layer (Fedus et al., 2022; Zoph et al., 2022). However, these constraints often sacrifice model flexibility and performance. Recent works reveal the potential performance degradation caused by such inflexible sparsity assignment (Huang et al., 2024; Liu et al., 2024). Moreover, to inspect the impact of such constraints, we plot the PPL-activation (PPL denotes perplexity) Pareto curve ($\text{activation ratio} = 1 - \text{sparsity ratio}$) of MoE in Figure 2 and compare it with a vanilla decoder-only Transformer (Touvron et al., 2023) of the same parameter

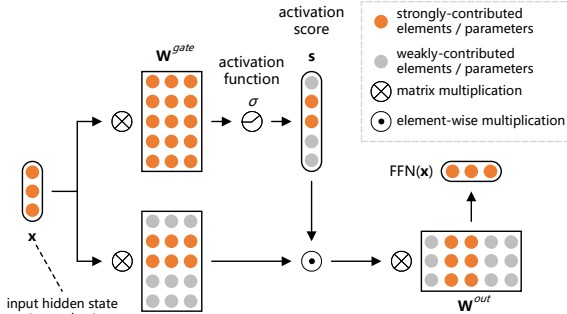 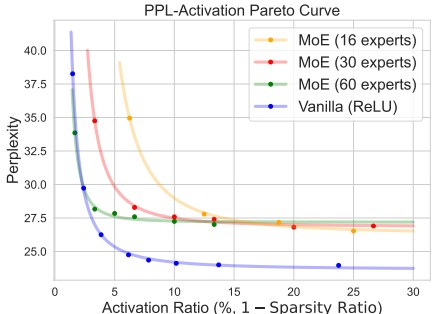

Figure 1: A typical case of activation sparsity (with a sparsity ratio of 60%) in a gated feed-forward network of LLMs, where considerable elements weakly contribute to the outputs within the activation scores.

Figure 2: The PPL-activation Pareto curve of the 0.1B MoE with different expert numbers versus the 0.1B vanilla decoder-only Transformer.

scale and amount of training data[1]. MoE has a significantly worse performance-sparsity trade-off. The best sparsity ratio is also hard to predefine, since a too-high or too-low sparsity ratio may lead to more severe performance degradation or substantial unnecessary computation, respectively.

To avoid negative impacts on flexibility and performance, we focus on the intrinsic activation sparsity within decoder-only Transformer-based LLMs in this paper, such as GPT (Brown et al., 2020) and LLaMA (Touvron et al., 2023). Different from the sparsity enforced through architectural constraints, intrinsic activation sparsity (Zhang et al., 2024a; Song et al., 2024a) arises self-adaptively during the pre-training stage, without unified or predefined sparsity ratios among different tokens or layers. We provide a comprehensive study on the quantitative scaling properties and influential factors of this intrinsic property. These findings can help us formulate a generalizable method to control and promote activation sparsity without harming flexibility and performance.

To rigorously study the influential factors of activation sparsity, we need a metric to precisely reflect the sparsity level of an LLM. Conventional works rely on prior experience to set a fixed global threshold for recognition of weakly-contributed neurons[2], which is empirical, inflexible, and performance-unaware (see Section 2.2). We propose an improved metric, named **PPL-$p$% sparsity**, with three advantages: versatility across model architectures, performance-awareness, and the precise recognition of weakly-contributed neurons. For versatility, PPL-$p$% sparsity follows Zhang et al. (2024a) and recognizes the most weakly-contributed neurons, which are then inactivated, by introducing layer-wise adaptive thresholds and comparing the magnitudes of neuron outputs with the layer-specific threshold. As a magnitude-based metric, it is not bound to the activation functions that output exactly non-negative elements (e.g., ReLU) and is applicable to any LLMs with activation layers. Moreover, considering the trade-off between performance and sparsity (Song et al., 2024a), performance-awareness is required, indicating whether a metric can comprehensively reflect the sparsity under each desired performance level. To this end, PPL-$p$% sparsity explicitly incorporates PPL as a performance proxy, computed as the ratio of inactivated neurons when the output PPL raises just by $p$% after the layer-wise thresholds are applied, compared to the dense setting with all neurons activated. Finally, the precise recognition of weakly-contributed neurons indicates that our metric obtains a good trade-off between performance and sparsity ratio (see Section 4.1).

Based on the above metric, we systematically study the correlation between the activation sparsity and influential factors, including **the amount of training data, the activation function, the width-depth ratio (i.e., the ratio of the hidden dimension to the layer number), and the parameter scale**. Through comprehensive experiments, we obtain the following observations:

1. There is an increasing power-law (SiLU-activated LLMs) or decreasing logspace power-law (ReLU-activated LLMs) relationship between the activation ratio and the amount of training data. Both laws are convergent with a certain limit sparsity ratio as the amount of

---

[1]MoE models of different sparsity are obtained by tuning the number of activated experts, while for the vanilla setting, we adjust the CETT value proposed by Zhang et al. (2024a).

[2]A neuron denotes a certain row or column within the parameter matrices.

data approaches infinity. Note that the increasing sparsity-data trend indicates that ReLU-activated LLMs are more efficient in improving activation sparsity with more data.

2. Given the same parameter scale, the sparsity obtained by ReLU-activated LLMs always surpasses that of SiLU-activated LLMs, while their performance is comparable.

3. Given the same parameter scale, the activation ratio linearly increases with the width-depth ratio under a bottleneck point (i.e., deeper models are sparser), above which the activation fluctuates around a fixed level. However, considering the performance issue, the best width-depth ratio should be just within a certain interval that ensures the best performance.

4. Given similar width-depth ratios, the limit of activation sparsity is weakly correlated to the scale of LLMs. On the other hand, the convergence speed to the limit is much faster in smaller models. We try to explain these phenomena in Section 4.4.

The above empirical laws can provide comprehensive instructional values for designing and pre-training an LLM with greater activation sparsity, which offers more significant potential in producing more efficient and interpretable LLMs. Moreover, our work enables the training-time prediction of the future sparsity ratio, and the evolving trend of activation sparsity with the amount of training data potentially provides a lens for the progress of neuron specialization.

## 2 PRELIMINARIES AND RELATED WORKS

### 2.1 PRELIMINARIES OF ACTIVATION SPARSITY

Activation sparsity is a prevalent property existing in neuron networks with activation layers, indicating the existence of considerable parameters, which correspond to the activation outputs with zero or low values and thus have a limited impact on final network outputs given specific inputs. These weakly-contributed parameters are often defined as "inactivated parameters" and can be skipped in computation. Due to the activation layers in feed-forward networks (FFNs), mainstream LLMs also present remarkable activation sparsity (Li et al., 2022; Zhang et al., 2022; Song et al., 2024a).

Owing to activation sparsity, we can improve LLMs in many aspects, such as efficiency, interpretability, and robustness. For instance, recent works manage to exploit activation sparsity for inference acceleration, mainly by predicting the activation patterns, wisely allocating hardware resources, and saving redundant resources associated with inactivated parameters (Liu et al., 2023; Song et al., 2023; Xue et al., 2024). Zhang et al. (2024b) reveal the existence of activation sparsity throughout the majority of the LLM pre-training process, and then utilize this sparsity for pre-training acceleration. Besides acceleration, activation sparsity also helps improve the interpretability (Agarap, 2018; Dai et al., 2022; Cuadros et al., 2022; Sajjad et al., 2022) and robustness (Ahmad & Scheinkman, 2019; Muthukumar & Sulam, 2023) of LLMs, which are also important properties in producing reliable and well-performing LLMs.

### 2.2 METRICS OF ACTIVATION SPARSITY

While considerable works call for greater activation sparsity due to the significant merits brought by sparsity (e.g., more efficient computing and better model interpretability) (Mirzadeh et al., 2023; Song et al., 2024a;b), it is nontrivial to frame a satisfactory metric for measuring sparsity.

For the convenience of demonstrations, we formally introduce the following notations for the computation process of FFNs (also see Figure 1). By denoting the hidden dimension and intermediate dimension as $d_h$ and $d_f$ respectively, a gated FFN (a common FFN form adopted in most mainstream LLMs (Dauphin et al., 2017; Shazeer, 2020)) processes the inputs as follows:

$$\mathbf{s} = \sigma(\mathbf{W}^{gate}\mathbf{x}), \quad \text{FFN}(\mathbf{x}) = \mathbf{W}^{out}[\mathbf{s} \odot (\mathbf{W}^{in}\mathbf{x})], \tag{1}$$

where $\mathbf{x} \in \mathbb{R}^{d_h}$, $\mathbf{s} \in \mathbb{R}^{d_f}$, $\sigma$, and $\odot$ denote the input hidden states, the activation scores, the activation function, and the element-wise multiplication, respectively. $\mathbf{W}^{gate}, \mathbf{W}^{in} \in \mathbb{R}^{d_f \times d_h}$ and $\mathbf{W}^{out} \in \mathbb{R}^{d_h \times d_f}$ are learnable parameters. Next, we decompose the parameters of FFN along the dimension of $d_f$ into $d_f$ neurons. The output of the $i$-th neuron $n_i$ is calculated by

$$s_i = \sigma(\mathbf{W}^{gate}_{i,:}\mathbf{x}), \quad n_i = \mathbf{W}^{out}_{:,i}[s_i \odot (\mathbf{W}^{in}_{i,:}\mathbf{x})], \quad \text{FFN}(\mathbf{x}) = \sum_{i=1}^{d_f} n_i, \tag{2}$$

where $\mathbf{W}_{i,:}^{gate}$, $\mathbf{W}_{i,:}^{in}$, $\mathbf{W}_{:,i}^{out}$ are the $i$-th row of $\mathbf{W}^{in}$, the $i$-th row of $\mathbf{W}^{gate}$, and the $i$-th column of $\mathbf{W}^{out}$, respectively. The FFN outputs can be formalized as the sum of all neuron outputs.

Activation sparsity is measured by the ratio of inactivated neurons, namely $|\mathcal{D}|/d_f$, where $\mathcal{D}$ is the index set of inactivated neurons. However, different sparsity metrics can differ in determining whether a specific neuron is inactivated. A straightforward metric, naturally adopted in ReLU-activated models, regards those neurons with zero activation scores in Eq. (2) as inactivated, namely $\mathcal{D} = \{i|s_i = 0\}$. To prune more non-zero weakly contributed activations, works by Kurtz et al. (2020) and Mirzadeh et al. (2023) introduce a positive threshold or bias. In this case, the set of inactivated neurons changes to $\mathcal{D} = \{i|s_i < \epsilon\}$, where $\epsilon > 0$ is the threshold or bias.

The major drawback of this straightforward definition is the lack of versatility. Concretely, it is unsuitable for activation functions that have considerable non-negligible negative outputs, such as SiLU (Elfwing et al., 2018). In these cases, the straightforward metric can lose considerable negative neuron outputs and harm performance. A quick fix is to use the absolute value, $\mathcal{D} = \{i||s_i| < \epsilon\}$, but a global threshold across layers is hard to determine. To this end, Zhang et al. (2024a) adaptively searches the layer-wise thresholds by introducing the cumulative errors of tail truncation (CETT). Defined as the $L_2$ norm relative error caused by inactivated neurons, CETT is computed as:

$$\text{CETT} = \frac{\|\sum_{i \in \mathcal{D}} n_i\|_2}{\|\text{FFN}(\mathbf{x})\|_2}, \quad \mathcal{D} = \{i|\|n_i\|_2 < \epsilon\}, \tag{3}$$

where $\|\cdot\|_2$ is the $L_2$ norm operator. Notably, as CETT increases monotonically with $\epsilon$, for each layer, we can use binary search to find a threshold $\epsilon$ that makes CETT just equal a predefined value.

Meeting the versatility demands, the CETT paradigm is still not friendly enough, as CETT does not directly reflect the model performance. In real-life deployment, we often need to utilize activation sparsity under a certain tolerance of performance degradation. Therefore, in this paper, we introduce a more performance-aware metric named PPL-$p\%$ sparsity, which explicitly reflects the sparsity ratio under a target PPL level by binary-searching an appropriate CETT value.

### 2.3 SCALING PROPERTIES OF ACTIVATION SPARSITY

Despite the importance of activation sparsity, few works conduct comprehensive studies on how it scales with the increase of parameter scale and training data, as well as the impact of other influential factors within architecture designs. Speculating that activation sparsity comes from the training dynamic in the optimization process, Li et al. (2022) finds an increasing trend of sparsity with larger scale, depth, and width in T5 series (Raffel et al., 2020). Zhang et al. (2024a) dives into this problem from the aspect of activation functions. Song et al. (2024a) discovers that LLMs tend to be sparser on more formatted datasets such as codes and multiple choices. Other works have discussed the scaling properties of parameter-sparse models (Frantar et al., 2023), fine-grained MoE models (Krajewski et al., 2024), and sparse autoencoders (Gao et al., 2024).

To the best of our knowledge, we present the first comprehensive quantitative study on the scaling properties of activation sparsity and the impact of architecture factors. The work most similar to ours is done by Li et al. (2022), but they only conduct qualitative analyses and focus on the conventional encoder-decoder Transformer (e.g., T5) rather than decoder-only LLMs. Besides, they adopt the most straightforward metric for activation sparsity, limited to ReLU-activated models.

## 3 METHODOLOGY

### 3.1 METRIC OF ACTIVATION SPARSITY

We propose **PPL-$p\%$ sparsity**, which improves the CETT metric (Zhang et al., 2024a) by directly reflecting the sparsity ratio at different requirements of PPL (as a proxy of performance). Firstly, given a pre-trained LLM, we introduce the dense setting with all its neurons activated (i.e., $|\mathcal{D}| = 0$), whose outputs are theoretically the most accurate. Next, we conduct a binary search for an appropriate CETT value $\text{CETT}_k$, which makes the average PPL increase just by $p\%$ compared to the dense setting. The search algorithm is described in Appendix C. Finally, based on $\text{CETT}_k$, we can determine the activation thresholds for each layer and obtain the sparsity value, according to

Eq. (3). Our experiments in Section 4.1 will demonstrate the rationality of this metric in achieving the greatest sparsity under the same output PPL.

## 3.2 INFLUENTIAL FACTORS

As we mainly study the effect of potentially influential factors on activation sparsity, it is necessary to specify how to compute the involved influential factors as follows:

- Amount of training data: The number of tokens passed during the pre-training process. To obtain more comprehensive dynamics and more precise estimations of the limits, the number of training tokens is no less than 80 times the scale of non-embedding parameters.
- Activation function: ReLU and SiLU are incorporated as the two most widely adopted activation functions with FFNs.
- Parameter scale: The number of parameters included in a model excluding the embeddings. In this work, we incorporate 5 scales: 0.1B, 0.2B, 0.4B, 0.8B, and 1.2B.
- Width-depth ratio: The ratio of the hidden dimension (i.e., $d_h$) to the number of hidden layers (denoted as $N$). We sample 9 values of width-depth ratio ranging from 14.2 to 597.3, covering both the ordinary and extreme conditions.

## 3.3 SETTING OF SCALING ANALYSIS

**Model settings**  We adopt the same architecture as MiniCPM (Hu et al., 2024), which adopts Tensor Program (Yang et al., 2022) for training stability and shares the input and output embeddings.

**Training settings**  We mainly focus on the activation sparsity of foundation models, which only undergo the pre-training stage. However, before we evaluate models on task-specific benchmarks, we follow MiniCPM (Hu et al., 2024) to conduct a decay stage, where instruction-tuning data is additionally added for training. Thereby, we can obtain more reasonable results on benchmarks. Besides, considering the influence of various training hyper-parameters, we follow the optimal batch sizes, optimal learning rates, and the WSD learning rate scheduler of MiniCPM (Hu et al., 2024).

**Evaluation Settings**  We introduce a tiny validation dataset and two groups of benchmarks, including commonsense reasoning and reading comprehension, for evaluation. We also test our model on more complex coding or knowledge benchmarks but fail to obtain a performance significantly surpassing the random level (See Appendix I), mainly due to the small parameter scales. For the measurement of sparsity, to eliminate the impact of stochastic factors (especially the sparsity fluctuations during the early stage), we use a sparsity stabilizing strategy to obtain smoother sparsity-data curves (see Appendix B for more details).

If there are no special statements, the training loss, validation loss, and perplexity are all calculated on models that only complete pre-training (the latter two metrics are computed on the validation data), and the task-specific performance is evaluated on checkpoints after the decay stage. See Appendix E and F for more details about datasets and training settings, respectively.

## 4 EXPERIMENTS

### 4.1 RATIONALITY OF PPL-$p\%$ SPARSITY

As stated in Section 1, a rational metric for activation sparsity should have: versatility across model architectures, performance-awareness, and the precise recognition of weakly-contributed neurons. While the former two properties are already demonstrated in Section 1, we mainly discuss whether PPL-$p\%$ sparsity has the third one in this section. Specifically, PPL-$p\%$ sparsity should wisely recognize more weakly-contributed neurons while minimizing performance degradation simultaneously. In other words, our metric should strike a better trade-off between performance and sparsity. We introduce the following baselines for comparison:

(1) **Straightforward ReLU**: The most simple setting that uses the zero threshold and is only applicable to ReLU: $\mathcal{D} = \{i | s_i = 0\}$.

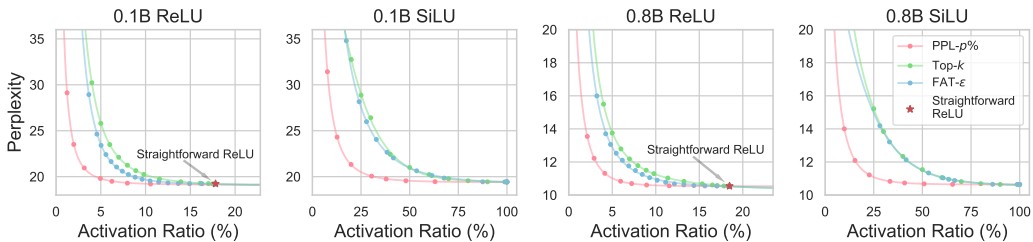

Figure 3: The PPL-activation Pareto curve of our PPL-$p\%$ sparsity versus two baselines within models of different scales. "Straightforward ReLU" is only applicable to ReLU-activated models.

Table 1: The average evaluation scores (%) on two task groups, where C.R. refers to *commonsense reasoning* and R.C. refers to *reading comprehension*. The second column represents settings with different $p\%$ values, with "dense" indicating the most accurate case where $p = 0$.

|  |  | 0.1B | | 0.2B | | 0.4B | | 0.8B | | 1.2B | |
|---|---|---|---|---|---|---|---|---|---|---|---|
|  |  | ReLU | SiLU | ReLU | SiLU | ReLU | SiLU | ReLU | SiLU | ReLU | SiLU |
| C.R. | dense | 49.6 | 49.5 | 52.0 | 52.2 | 54.7 | 55.8 | 56.8 | 57.6 | 60.0 | 59.6 |
|  | PPL-1% | 49.1 | 49.9 | 51.7 | 52.4 | 54.6 | 55.8 | 55.9 | 57.6 | 59.6 | 59.6 |
|  | PPL-5% | 49.2 | 49.0 | 51.7 | 52.0 | 54.3 | 55.1 | 56.3 | 57.1 | 59.3 | 58.8 |
|  | PPL-10% | 49.4 | 48.7 | 51.6 | 51.9 | 54.9 | 55.2 | 55.6 | 56.4 | 59.3 | 59.3 |
| R.C. | dense | 28.2 | 27.7 | 40.7 | 40.2 | 44.0 | 41.8 | 44.8 | 43.3 | 53.2 | 54.8 |
|  | PPL-1% | 28.4 | 28.0 | 39.7 | 39.6 | 42.9 | 40.9 | 43.2 | 44.3 | 53.3 | 55.4 |
|  | PPL-5% | 26.9 | 26.5 | 38.6 | 36.8 | 40.8 | 38.2 | 42.2 | 40.7 | 53.3 | 52.6 |
|  | PPL-10% | 26.2 | 24.8 | 38.6 | 34.4 | 39.9 | 35.3 | 40.3 | 38.8 | 52.9 | 51.1 |

(2) **Top-$k$ sparsity**, widely adopted in the MoE architectures (Fedus et al., 2022; He, 2024), enforces each layer to consistently maintain $k$ activated neurons, whose absolute values of activation scores rank in the top-k ones among all the neurons of that layer. Obviously, we have $|\mathcal{D}| = d_f - k$, and the Top-$k$ sparsity method holds a constant sparsity ratio across all layers.

(3) **FAT-$\epsilon$ sparsity** (Kurtz et al., 2020) (FAT denotes forced activation threshold) similarly introduces a global hyper-parameter $\epsilon$ as the threshold shared by all layers, namely $\mathcal{D} = \{i||s_i| < \epsilon\}$. Note that this is slightly different from the original FATReLU by Kurtz et al. (2020) as we compute the absolute values of activation scores to accommodate SiLU.

As demonstrated by Figure 3, PPL-$p\%$ sparsity obtains the best trade-off between sparsity and performance, with the lowest PPL given a target sparsity ratio. This is brought by its wise recognition of inactivated neurons through adaptive thresholds. Moreover, the performance-aware definition of PPL-$p\%$ sparsity makes us explicitly grasp the sparsity ratio at each desired perplexity. The advantage of PPL-$p\%$ also holds when considering the downstream task performance. (See Appendix G)

Another problem lies in how the hyper-parameter $p\%$ influences task-specific performance. To inspect this, we evaluate models of different scales on the benchmarks in Section 3.3 and tune their sparsity through distinct $p\%$ values. As shown in Table 1, with the value of $p\%$ increasing (intrinsically promoting greater sparsity), the reading comprehension performance is considerably impaired, corresponding to the trade-off between sparsity and performance. Notably, in both task groups, the average performance of "PPL-1%" is comparable to that of the theoretically most accurate "dense" setting. Therefore, **we assume PPL-1% sparsity as a reliable performance-unimpaired metric and employ it to compute sparsity in the following discussions**.

### 4.2 ACTIVATION SPARSITY WITH THE AMOUNT OF TRAINING DATA AND ACTIVATION FUNCTIONS

To obtain the scaling relationship between the activation sparsity and the amount of training data, we pre-train models with different numbers of parameters and two activation functions (i.e., ReLU and SiLU), respectively, and then evaluate the sparsity level of their checkpoints using PPL-1%. After careful attempts, we find that the curve of activation ratios to the amount of training data is easier

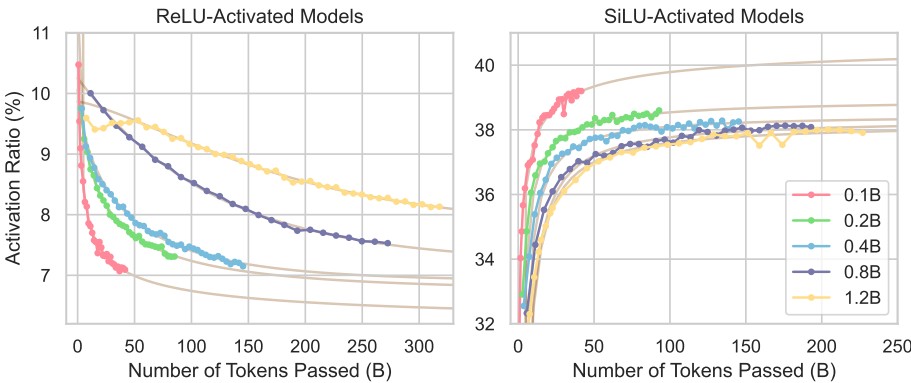

Figure 4: The trend of activation ratios (hereinafter using PPL-1% sparsity) of models with different scales and activation functions during the pre-training stage. The fitted curves are plotted in brown. The number of training tokens is no less than 190 times the scale of non-embedding parameters.

to fit than that of the sparsity ratio. Therefore, we will frequently use activation ratios instead of sparsity ratios in the following sections, whose trend is plotted in Figure 4.

For ReLU models, we observe a logspace power-law relationship between the activation ratio $A_{ReLU}(D)$ and the amount of training data $D$, expressed in the following formula:

$$A_{ReLU}(D) = \exp(-cD^\alpha + b) + A_0, \tag{4}$$

where $A_0 > 0$ is the limit activation ratio with infinite training data and we have $c, \alpha > 0$. Obviously, this is a convergent decreasing function, indicating that **more training data can potentially make ReLU models more sparsely activated**.

By contrast, the activation ratio $A_{SiLU}(D)$ of SiLU models exhibit a vanilla power-law relationship:

$$A_{SiLU}(D) = -\frac{c}{D^\alpha} + A_0, \tag{5}$$

where similarly, $A_0 > 0$ is the limit activation ratio and $c, \alpha > 0$. Note that this is a convergent increasing function, and thus **more training data will impair the activation sparsity of SiLU models**. See Appendix D for the algorithm of curve fitting and the results (i.e., coefficients).

As for the selection of activation functions, by comparing the sparsity dynamics, we can conclude that the activation sparsity achieved by ReLU is significantly greater than that of SiLU. Besides, the task-specific performance in Table 1 and the trend of training loss in Appendix A reveal the comparable performance between ReLU and SiLU activations. Based on the above observations, **ReLU is more competent as the activation function than SiLU due to three advantages**: an increasing trend of sparsity, significantly higher sparsity ratio, and comparable performance.

## 4.3 ACTIVATION SPARSITY WITH THE WIDTH-DEPTH RATIO

The width-depth ratio, defined as the ratio of the hidden dimension to the layer number, reflects the shape of a Transformer and is a key architectural property that potentially influences activation sparsity. To inspect its influence on the activation sparsity, we conduct experiments on the 0.1B ReLU-activated model and select 9 different width-depth ratios. The limit activation ratio and the limit training loss are plotted in Figure 5 and Figure 6 respectively.

As demonstrated by Figure 5, **under a bottleneck point (about 114 for 0.1B), the activation ratio linearly increases with the width-depth ratio**. However, given a width-depth ratio greater than this bottleneck, the activation ratio fluctuates around 8%. From the sparsity aspect, a smaller width-depth ratio is definitely more helpful. However, the dynamics shown in Figure 6 demonstrate that there exists the best interval of width-depth ratio for the lowest training loss (from 74 to 282 for 0.1B). Therefore, to maintain the best performance while promoting greater activation sparsity, the best width-depth ratio should fall on the smallest point of this interval (i.e., around 74 for 0.1B).

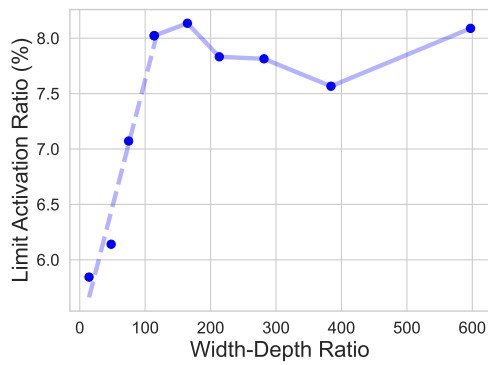

Figure 5: The limit activation ratios on 0.1B ReLU-activated models.

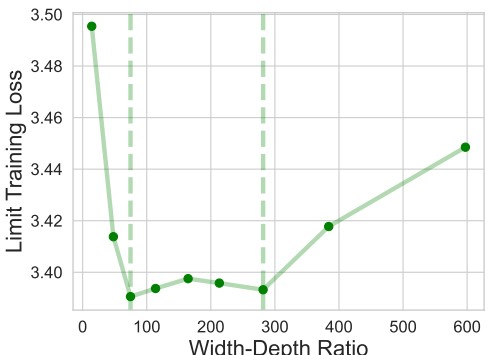

Figure 6: The limit training loss on 0.1B ReLU-activated models.

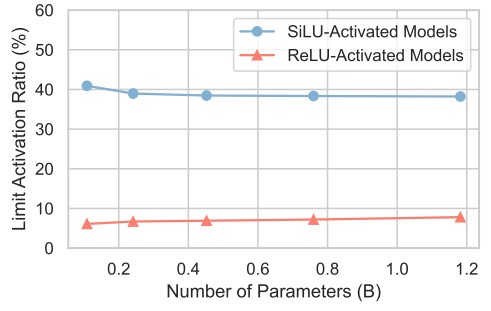

Figure 7: The limit activation ratio for pre-trained models with different scales and activation functions.

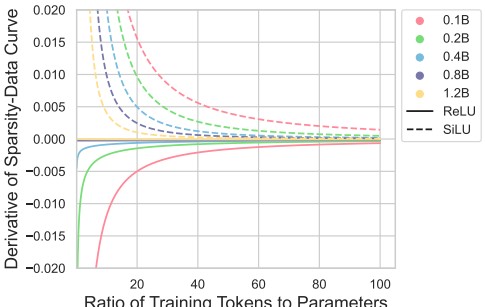

Figure 8: The derivative trends of the sparsity-data curve with the increase of data-scale ratio, within ReLU/SiLU models of distinct scales.

## 4.4 ACTIVATION SPARSITY WITH THE PARAMETER SCALE

To obtain comprehensive scaling properties of activation sparsity with the increase of scales (i.e., the number of non-embedding parameters), we obtain the limit activation ratio of the above pre-trained models with 5 distinct scales but similar width-depth ratios. From the results plotted in Figure 7, we can reach the first observation that **under similar width-depth ratios, the limit activation ratio as the amount of training data approaches infinity is weakly related to the parameter scale**. For SiLU settings, the activation ratio decreases slightly by 2.7 points from 0.1B to 1.2B. By contrast, for ReLU settings, the activation ratio marginally increases by 1.7 points from 0.1B to 1.2B.

To reflect the evolving dynamics of sparsity, we compute the derivatives of the sparsity-data curve as fitted in Section 4.2 and plot the trend of derivatives with the increase of data-scale ratio[3]. The results in Figure 8 clearly demonstrate that **smaller models tend to converge faster than larger models to the limit**, as the absolute values of their derivatives are much larger. We try to explain the above observations as follows.

**Observation: Neurons within models of different scales present similar activation patterns.** To support this point, we conduct two experiments from the aspect of *dataset-wise* and *token-wise* activation patterns respectively.

To inspect the dataset-wise activation distributions under different parameter scales, we consider four datasets, each a subset of the pre-training data: Code, Wikipedia, Math, and Chinese. Next, we compute the distribution of activation frequencies (i.e., the times that a neuron is activated divided by

---

[3]The data-scale ratio means the ratio of the number of training tokens to the parameter scale. We choose this variable as previous works have demonstrated the roughly proportional relationship between the optimal amount of training data and the parameter scale (Hoffmann et al., 2022; Besiroglu et al., 2024).

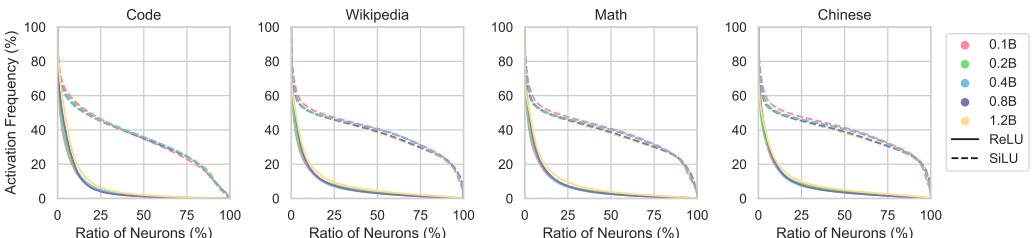

Figure 9: The distribution of the neuron activation frequencies within models of distinct scales. Four datasets from the pre-training data are involved.

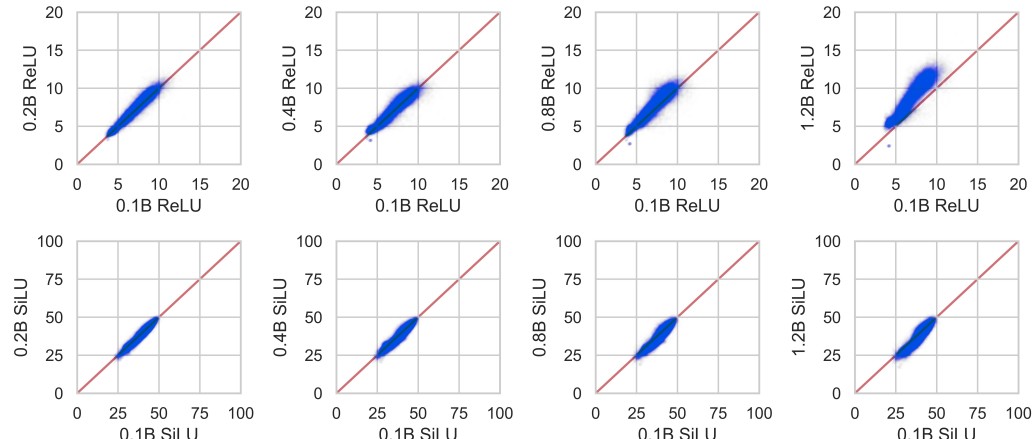

Figure 10: The activation ratio (%) distributions of 71,549 tokens sampled from the vocabulary. We conduct a pair-wise comparison of the average activation ratio of each token within models of different scales. Note that the red line is the $y = x$ curve.

the total number of tokens) among the neurons within models of different scales. As demonstrated by Figure 9, for all the datasets, the distribution patterns of neuron activation frequencies are similar across different scales. While this observation holds on average, special cases exist in certain layers (see Appendix H).

For the token-wise activation, we sample 71,549 tokens from the vocabulary and count their activation ratios on a sufficiently large amount of data. Next, we compare the activation ratios of each token among models of different scales in a pair-wise manner. Figure 10 clearly shows that most tokens maintain a close activation ratio across models of various scales.

From different aspects, the above two experiments both support the insensitiveness of the activation pattern to the parameter scale. This can potentially provide one explanation for why the activation sparsity is quite weakly correlated with the model sizes.

**Assumption: Neurons within models of different scales present similar specialization.** The neurons in FFNs tend to specialize into certain functions during the training progress (Li et al., 2022; Zhang et al., 2023). However, few works have studied how such specialization differs in models of distinct scales. As stated above, both the dataset-wise and token-wise activation patterns are insensitive to the parameter scale. In other words, the numerical distribution of neurons activated for a certain function (e.g., a specific category of datasets or syntactic elements) is similar. Therefore, it is reasonable to assume that the specialization of neurons is also scale-insensitive.

**Deduction: Smaller models converge faster to the limit activation ratio mainly due to their small amount of neurons.** To simplify this problem, we model the specialization of neurons as a grouping process, where each neuron can be placed into zero or more groups (considering the potential existence of dead neurons and versatile neurons). Suppose the $d_f$ neurons should specialize into $G$ groups, each of them having $t_1, t_2, ..., t_G$ neurons respectively. Based on the assumption of

similar activation patterns and neuron specialization, the ratio of neurons placed in each group (i.e., $0 < t_i/d_f \leq 1,\ i = 1, 2, ..., G$) should be shared across different parameter scales. We can obtain the number of all the possible grouping results $T(d_f)$ easily,

$$T(d_f) = \prod_{i=1}^{G} C_{d_f}^{t_i} = \prod_{i=1}^{G} \frac{d_f!}{t_i!(d_f - t_i)!}, \qquad (6)$$

where $C_{d_f}^{t_i}$ is the combinatorial number, the number of possibilities to select $t_i$ neurons from $d_f$ ones. Obviously, $T(d_f)$ grows in a factorial speed with $d_f$, much faster than the linear function. For larger models, the number of neuron specialization possibilities is significantly greater than that of smaller ones. Therefore, it takes more training expenses for larger models to form stable neuron specialization and approach the limit activation ratio.

## 5 DISCUSSION

In this section, we mainly discuss how this work can help accomplish our key target: *how to obtain an LLM with greater activation sparsity*.

**Architectural design**     Before kicking off the pre-training stage, a wiser design of the LLM architecture can effectively improve the limit activation sparsity. Based on the discoveries in this paper, we propose to replace SiLU with ReLU as the activation function to promote greater sparsity and leverage its increasing trend of sparsity with more training data. Moreover, a deeper model can potentially exhibit higher sparsity given a fixed computation budget. Nevertheless, as a model with an extreme width-depth ratio can violate the performance, the best setting for the width and depth still deserves careful studies, combined with the quantitative study on the relationship between performance and the width-depth ratio.

**Training-time predictable sparsity**     Another important value of our work lies in the prediction of sparsity during the pre-training stage. By fitting the power-law or logspace power-law between activation sparsity and the amount of training data, model developers can either predict the theoretical upper/lower bound sparsity of a model to evaluate its potential (e.g., in inference acceleration), or estimate the number of tokens required to achieve a desired sparsity ratio.

**Lens for the convergence of neuron specialization**     Generally, the loss curve is one of the most important signs of the training state (e.g., at which point the model converges). However, if we compare the loss curve in Figure 11 and the sparsity (activation ratio) curve in Figure 4, we will find that **the convergence speed of activation sparsity is much slower than loss, indicating the ongoing of neuron specialization even when the training loss changes little**. Despite the wide recognition of the neuron specialization phenomenon (Li et al., 2022; Zhang et al., 2023), it is still unclear when such specialization converges and how to inspect this progress. Besides, the loss curve is often not a good standard for convergence, especially for modern LLMs with trillion-level pre-training data. We believe that the trend of **activation sparsity can provide a lens for inspecting the progress of neuron specialization as well as the training convergence**. We leave the study on the correlation between activation sparsity and neuron specialization for future work.

## 6 CONCLUSION

In this work, we first propose a precise, versatile, and performance-aware metric for activation sparsity, called PPL-$p\%$ sparsity, whose rationality is demonstrated in striking a better trade-off between performance and sparsity. Next, we conduct a comprehensive and quantitative study on how the activation sparsity scales with the amount of training data and parameter scales. The influence of two key architectural settings, namely the activation function and the width-depth ratio, are also well evaluated. Through extensive experiments, we figure out the quantitative sparsity-data relationship, substantiate the advantage of ReLU activation, find a small width-depth ratio helpful in promoting sparsity, observe and then explain the insensitiveness of sparsity to scale. These can better instruct LLM developers to build models with greater activation sparsity and leverage the merits of sparsity.

## LIMITATIONS

One limitation lies in the lack of experiments on even larger LLMs, such as models with more than 7B parameters. Such experiments will cost considerable computation resources and time but may potentially encounter unexpected observations (e.g., emergent properties).

Another drawback of our study is the absence of computation (e.g., FLOPS) in some analyses, especially the experiments for width-depth ratios. In Section 4.3, we find a smaller width-depth ratio potentially produces a sparser model. However, with the substantial increase in the number of layers, the training efficiency is significantly decreased, as we have observed in the training process. Therefore, in addition to performance, the computation costs of a model also deserve consideration. Similarly, considering the values of activation sparsity in acceleration, it may be interesting to involve the inference computation as a variable in our study.

Finally, an obvious limitation of our PPL-$p\%$ metric (as well as all the sparsity metrics relying on a validation dataset) is the sensitivity to different data distributions. Intuitively, the same model can have different sparsity ratios on distinct datasets or tasks. The correlation between sparsity and influential factors (e.g., the form of power laws) can also have dataset-unique characteristics. A piece of evidence already presented in our paper is in Table 1, where the performance on commonsense reasoning tasks is insensitive to $p\%$, largely different from the results on reading comprehension tasks. Moreover, the data mixing policies for pre-training can also have a considerable impact on the activation sparsity, which we leave for future work.

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

## A    TRAINING LOSS DYNAMICS

To present the comprehensive training dynamics of our pre-trained models, we plot the trend of loss with the increase of training data in Figure 11. As can be clearly observed, larger models have smaller training loss. Besides, we also plot the limit values of the training loss with infinite training tokens, shown in Figure 12. As demonstrated in the above two figures, SiLU and ReLU models are well comparable from the loss aspect.

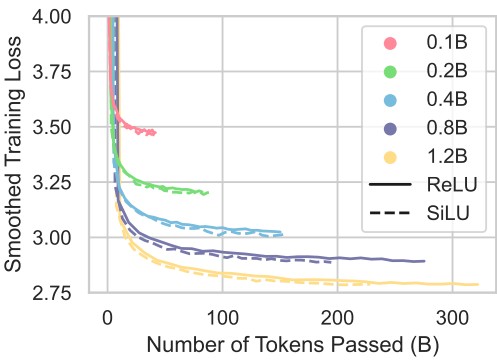

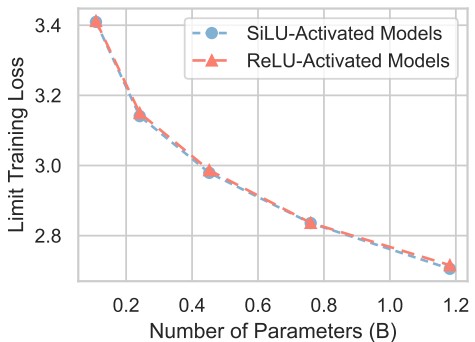

Figure 11: The trend of pre-training loss for models with different scales and activations.

Figure 12: The limits of the training loss with the amount of training data approaches infinity.

## B    SPARSITY STABILIZING STRATEGY

We find that the stochastic factors in gradient descent during pre-training have a significant impact on the metric of activation sparsity. Especially, during the early training stage, the model is far from convergence with considerable sparsity fluctuations, and the magnitude-based sparsity metric can become unstable, making the sparsity-data curve not smooth enough.

To eliminate the influence of these unstable factors to facilitate smoother sparsity metric, we first drop the sparsity points during the warmup stage for curve fitting. Moreover, we mainly apply the PPL-$p\%$ on the last several checkpoints (specifically, the last five pre-trained checkpoints) as a whole, binary-searching a CETT value that controls the average PPL of these checkpoints to just increase by $p\%$. Then this CETT value is applied to all the checkpoints of this pre-training process to measure the sparsity.

## C    BINARY SEARCH ALGORITHM FOR CETT

Given a list of checkpoints, a validation dataset, and a hyper-parameter $p\%$, we employ Algorithm 1 to find the CETT value that just makes the average PPL of these checkpoints on the validation dataset rise by exactly $p\%$, compared to the dense setting with all the neurons activated. The rationality of using binary search is substantiated by the monotonous relationship between PPL and CETT, as shown in Figure 13. Note that this algorithm can be applied to either a single checkpoint or multiple checkpoints, as adopted in the strategy described in Appendix B.

## D    FITTING ALGORITHM AND RESULTS

We employ the Levenberg-Marquardt method (Marquardt, 1963) to fit the activation-data curves. To improve the stability of curve fitting, we divide the number of tokens passed (i.e., the amount of training data) by $10^9$ to normalize its magnitude. All the results we obtained from fitting Eq. (4) (for ReLU-activated models) and Eq. (5) (for SiLU-activated models) are shown in Table 2.

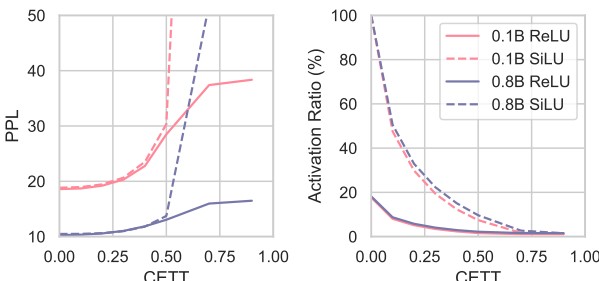

Figure 13: Experiments show that both PPL and the activation ratio change monotonously with CETT, the key hyper-parameter of PPL-$p\%$.

---

**Algorithm 1** Find the CETT value for PPL-$p\%$ sparsity

---

**Input:** The input list of checkpoints $CkptList$.
**Input:** The validation dataset $ValidSet$.
**Input:** The hyper-parameter $p\%$.
**Input:** The error tolerance $eps$.
**Output:** The $CETT$ that just makes the average PPL of $CkptList$ on $ValidSet$ rise by $p\%$.
$l \leftarrow 0,\ r \leftarrow 1$
**while** $r - l > eps$ **do**
  $mid \leftarrow (l + r)/2$
  $PPLRatioList \leftarrow [\ ]$
  **for** $Ckpt \in CkptList$ **do**
    $loss_{dense} \leftarrow L_{dense}(Ckpt, ValidSet)$
    $loss_{sparse} \leftarrow L_{sparse}(Ckpt, ValidSet, cett = mid)$
    $PPLRatio \leftarrow \exp(loss_{sparse} - loss_{dense})$
    $PPLRatioList.append(PPLRatio)$
  **end for**
  $MeanPPLRatio \leftarrow \mathrm{Mean}(PPLRatioList)$
  **if** $MeanPPLRatio < 1 + p\%$ **then**
    $l \leftarrow mid$
  **else**
    $r \leftarrow mid$
  **end if**
**end while**
$CETT \leftarrow (l + r)/2$
**return** $CETT$

---

## E DATASETS AND BENCHMARKS

**Training data** The pre-training data is a mixture of various corpus, including a cleaned version of CommonCrawl, Dolma (Soldaini et al., 2024), C4 (Raffel et al., 2020), Pile (Gao et al., 2020), the Stack (Kocetkov et al., 2022), StarCoder (Li et al., 2023), and other collected raw corpus. In contrast, the decay data contains additional instruction-tuning data, such as UltraChat (Ding et al., 2023), SlimOrca (Colombo et al., 2024), OssInstruct (Wei et al., 2024), EvolInstruct (Xu et al., 2023), and other collected datasets.

**Validation data** To measure the PPL-$p\%$ sparsity more precisely, we introduce a tiny validation dataset, which shares the same distribution as the pre-training data. We conduct deduplication to eliminate any intersections between validation and pre-training data.

**Evaluation benchmarks** To evaluate the task-specific performance of models, we introduce the following two groups of benchmarks: (1) *Commonsense reasoning*: We compute the average 0-shot accuracies on PIQA (Bisk et al., 2020), SIQA (Sap et al., 2019), HellaSwag (Zellers et al., 2019),

Table 2: Coefficients of activation-data (logspace) power-laws obtained from curve fitting. The curves of ReLU-activated and SiLU-activated models follow Eq. (4) and Eq. (5) respectively.

| | | $\alpha$ | $b$ | $c$ | $A_0$ |
|---|---|---|---|---|---|
| ReLU | 0.1B | $1.01 \times 10^{-01}$ | $-1.51 \times 10^{-02}$ | $3.20 \times 10^{+00}$ | $6.14 \times 10^{-02}$ |
| | 0.2B | $4.49 \times 10^{-01}$ | $-3.05 \times 10^{+00}$ | $2.86 \times 10^{-01}$ | $6.74 \times 10^{-02}$ |
| | 0.4B | $6.83 \times 10^{-01}$ | $-3.46 \times 10^{+00}$ | $7.90 \times 10^{-02}$ | $6.90 \times 10^{-02}$ |
| | 0.8B | $1.01 \times 10^{+00}$ | $-3.49 \times 10^{+00}$ | $7.97 \times 10^{-03}$ | $7.21 \times 10^{-02}$ |
| | 1.2B | $1.33 \times 10^{+00}$ | $-3.89 \times 10^{+00}$ | $9.03 \times 10^{-04}$ | $7.82 \times 10^{-02}$ |
| SiLU | 0.1B | $4.79 \times 10^{-01}$ | - | $1.02 \times 10^{-01}$ | $4.09 \times 10^{-01}$ |
| | 0.2B | $8.44 \times 10^{-01}$ | - | $2.08 \times 10^{-01}$ | $3.90 \times 10^{-01}$ |
| | 0.4B | $1.03 \times 10^{+00}$ | - | $4.20 \times 10^{-01}$ | $3.85 \times 10^{-01}$ |
| | 0.8B | $9.95 \times 10^{-01}$ | - | $5.62 \times 10^{-01}$ | $3.83 \times 10^{-01}$ |
| | 1.2B | $9.67 \times 10^{-01}$ | - | $5.38 \times 10^{-01}$ | $3.82 \times 10^{-01}$ |

WinoGrande (Sakaguchi et al., 2020), and COPA (Roemmele et al., 2011). (2) *Reading comprehension*: We report the average 0-shot accuracies on BoolQ (Clark et al., 2019), LAMBADA (Paperno et al., 2016), and TyDi QA (Clark et al., 2020).

We also evaluate our model on more complex tasks but fail to obtain performance above the random level. These include: the average pass@1 scores on HumanEval (0-shot) (Chen et al., 2021) and MBPP (3-shot) (Austin et al., 2021), the average accuracies on GSM8K (8-shot) (Cobbe et al., 2021), MMLU (5-shot) (Hendrycks et al., 2020), Big Bench Hard (BBH) (3-shot) (Suzgun et al., 2022), and AGI-Eval (0-shot) (Zhong et al., 2023).

## F DETAILED TRAINING SETTINGS

We utilize the MiniCPM (Hu et al., 2024) architecture and adopt its hyper-parameter policies, along with the WSD learning rate scheduling method. Across all parameter scales, the ratio of $d_f$ to $d_h$ is equal to 2.5 consistently, the number of query heads always matches that of key and value heads, and the width-depth ratios range from 48 to 56, generally similar across different scales. The specific number of parameters of various settings are shown in Table 3. We employ the following pre-training hyper-parameters across all settings: peak learning rate $lr = 0.01$, $\beta_1 = 0.9$, $\beta_2 = 0.95$, $weight\ decay = 0.1$. The batch size depends on the parameter scale, as presented in Table 3.

Table 3: Hyper-parameters across various parameter scales.

| Parameter Scale | 0.1B | 0.2B | 0.4B | 0.8B | 1.2B |
|---|---|---|---|---|---|
| # non-embedding parameters | $1.08 \times 10^8$ | $2.41 \times 10^8$ | $4.52 \times 10^8$ | $7.60 \times 10^8$ | $1.18 \times 10^9$ |
| batch size | $3.27 \times 10^5$ | $5.90 \times 10^5$ | $7.86 \times 10^5$ | $1.18 \times 10^6$ | $1.57 \times 10^6$ |

## G TRADE-OFF BETWEEN SPARSITY AND DOWNSTREAM TASK PERFORMANCE

In Section 4.1, we have demonstrated that PPL-$p\%$ obtains the best trade-off between PPL and sparsity. However, concerns may arise that lower PPL does not necessarily present better performance on specific downstream tasks. Although many existing works have revealed the statistically monotonous correlation between PPL and task performance (Owen, 2024; Dubey et al., 2024), to consolidate our work, we still provide detailed experiments on the trade-off between task performance and sparsity. Using different sparsity metrics, the Pareto curves are shown in Figure 14.

From the above results, we can demonstrate that PPL-$p\%$ still obtains the best trade-off between downstream task performance and sparsity. Specifically, with the increase of sparsity, both Top-$k$

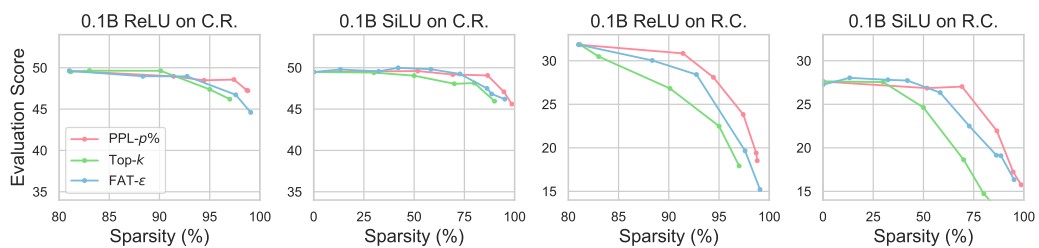

Figure 14: The Pareto curves between downstream task performance (i.e., evaluation score on benchmarks) and activation sparsity using different sparsity metrics.

and FAT-$\epsilon$ start to show significant performance degradation (i.e., decrease by more than 5%) at a sparsity point consistently lower than PPL-$p\%$. On minor data points with low sparsity levels, two baselines may slightly surpass PPL-$p\%$, but the performance gaps are no more than 1 point.

## H  DATASET-WISE ACTIVATION PATTERN

Although the overall distribution patterns of activation frequencies are similar in terms of the average scenario, they exhibit difference when focusing on neurons in specific layers, such as the first, the last, or the exact middle layer. As shown in Figure 15, models with varying parameter scales have diverse neuron activation frequency distributions in the first layer and the last layer, while the patterns on the middle layer are still largely scale-insensitive.

## I  PERFORMANCE ON INDEPENDENT BENCHMARKS

In Table 1, we already provide the average performance on the two groups of commonsense reasoning and reading comprehension. In this section, we present the evaluation scores on independent benchmarks of these two task groups, as shown in Table 4 and Table 5, respectively. From these tables, it can be observed that in commonsense reasoning benchmarks, as the number of model parameters increases from 0.1B to 1.2B, the average evaluation score of the ReLU settings rises from 49.6 to 60.0, while the average score of the SiLU settings increases from 49.5 to 59.6. Similarly, in reading comprehension benchmarks, the score of ReLU settings goes from 28.2 to 53.2, and the score of SiLU settings rises from 27.7 to 54.8. Additionally, models with these two distinct activation functions demonstrate comparable performance at the same parameter scale. Moreover, under the PPL-1% setting, the models are generally comparable to the dense setting with all neurons activated, whereas under the PPL-5% setting, they tend to suffer from significant performance on reading comprehension tasks, but the commonsense reasoning scores almost remain unaffected, which is a phenomenon worth studies.

We also evaluate our models on several more complex tasks. However, due to the limited number of parameters, we are unable to obtain reliable results above the random level. The evaluation results for this part are shown in Table 6.

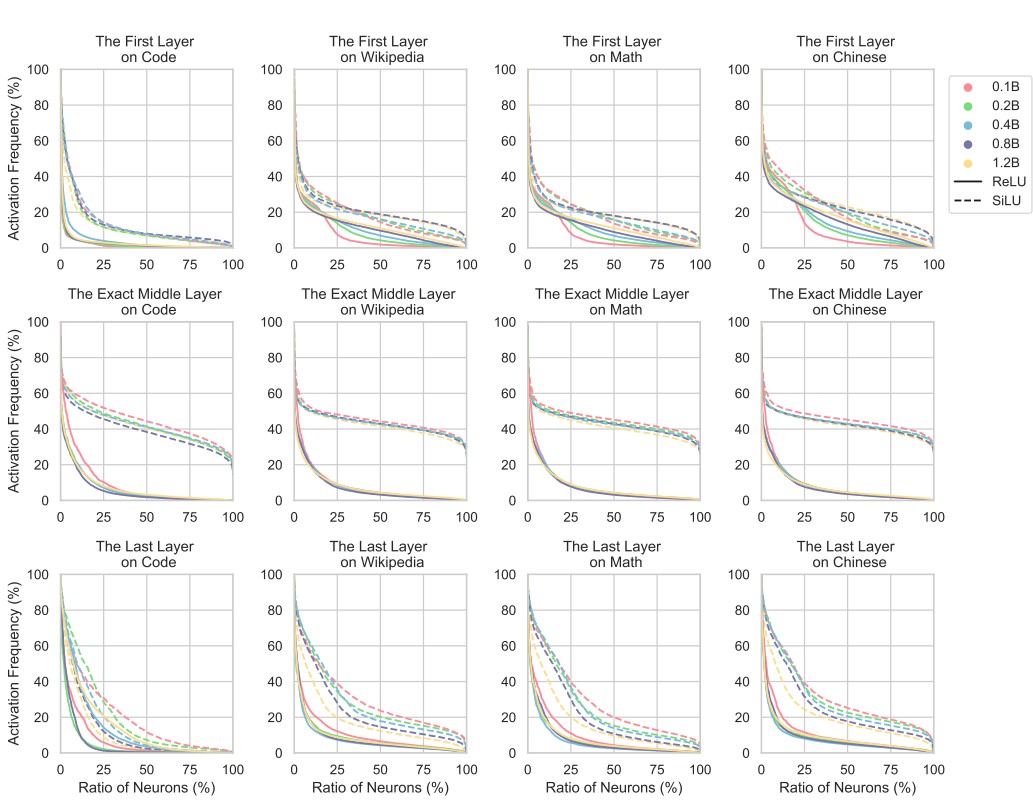

Figure 15: The distributions of average activation frequencies across three individual layers at different positions within models of distinct scales, including four datasets from the pre-training data.

Table 4: Evaluation scores (%) on *commonsense reasoning* benchmarks.

| | | | PIQA | SIQA | HellaSwag | WinoGrande | COPA | Avg. |
|---|---|---|---|---|---|---|---|---|
| | | | acc | acc | acc | acc | acc | |
| 0.1B | ReLU | dense | 62.8 | 37.8 | 30.5 | 53.0 | 64.0 | 49.6 |
| | | PPL-1% | 62.7 | 37.4 | 30.5 | 52.6 | 62.0 | 49.1 |
| | | PPL-5% | 63.1 | 37.6 | 30.3 | 51.1 | 64.0 | 49.2 |
| | | PPL-10% | 63.0 | 38.0 | 30.5 | 51.5 | 64.0 | 49.4 |
| | SiLU | dense | 64.3 | 37.6 | 30.9 | 52.8 | 62.0 | 49.5 |
| | | PPL-1% | 64.3 | 37.5 | 30.7 | 53.0 | 64.0 | 49.9 |
| | | PPL-5% | 63.5 | 38.4 | 30.5 | 51.5 | 61.0 | 49.0 |
| | | PPL-10% | 63.8 | 38.1 | 30.4 | 51.3 | 60.0 | 48.7 |
| 0.2B | ReLU | dense | 66.3 | 38.3 | 37.1 | 53.1 | 65.0 | 52.0 |
| | | PPL-1% | 66.3 | 38.1 | 37.2 | 52.7 | 64.0 | 51.7 |
| | | PPL-5% | 66.2 | 38.1 | 37.1 | 52.2 | 65.0 | 51.7 |
| | | PPL-10% | 66.0 | 37.9 | 37.0 | 51.9 | 65.0 | 51.6 |
| | SiLU | dense | 67.6 | 39.0 | 37.8 | 51.8 | 65.0 | 52.2 |
| | | PPL-1% | 68.2 | 39.2 | 37.7 | 52.0 | 65.0 | 52.4 |
| | | PPL-5% | 67.4 | 38.2 | 37.7 | 51.8 | 65.0 | 52.0 |
| | | PPL-10% | 66.8 | 38.8 | 37.9 | 52.1 | 64.0 | 51.9 |
| 0.4B | ReLU | dense | 68.8 | 39.9 | 42.7 | 51.9 | 70.0 | 54.7 |
| | | PPL-1% | 68.8 | 39.7 | 42.9 | 51.8 | 70.0 | 54.6 |
| | | PPL-5% | 68.3 | 39.9 | 42.7 | 52.5 | 68.0 | 54.3 |
| | | PPL-10% | 68.1 | 40.4 | 42.6 | 53.2 | 70.0 | 54.9 |
| | SiLU | dense | 69.0 | 39.6 | 44.5 | 51.9 | 74.0 | 55.8 |
| | | PPL-1% | 68.7 | 39.4 | 44.6 | 52.2 | 74.0 | 55.8 |
| | | PPL-5% | 68.9 | 39.4 | 44.6 | 51.5 | 71.0 | 55.1 |
| | | PPL-10% | 68.7 | 39.3 | 44.9 | 51.0 | 72.0 | 55.2 |
| 0.8B | ReLU | dense | 70.1 | 41.8 | 50.4 | 53.6 | 68.0 | 56.8 |
| | | PPL-1% | 69.8 | 41.8 | 50.2 | 52.8 | 65.0 | 55.9 |
| | | PPL-5% | 69.9 | 41.8 | 49.7 | 52.3 | 68.0 | 56.3 |
| | | PPL-10% | 69.6 | 41.8 | 50.0 | 51.8 | 65.0 | 55.6 |
| | SiLU | dense | 70.4 | 40.9 | 50.6 | 54.0 | 72.0 | 57.6 |
| | | PPL-1% | 70.3 | 41.4 | 50.6 | 53.9 | 72.0 | 57.6 |
| | | PPL-5% | 69.9 | 41.3 | 51.0 | 54.1 | 69.0 | 57.1 |
| | | PPL-10% | 69.5 | 40.7 | 50.6 | 53.2 | 68.0 | 56.4 |
| 1.2B | ReLU | dense | 71.6 | 44.1 | 57.7 | 56.4 | 70.0 | 60.0 |
| | | PPL-1% | 71.1 | 44.7 | 58.0 | 55.3 | 69.0 | 59.6 |
| | | PPL-5% | 70.8 | 43.9 | 57.8 | 54.9 | 69.0 | 59.3 |
| | | PPL-10% | 70.2 | 43.6 | 57.1 | 53.7 | 72.0 | 59.3 |
| | SiLU | dense | 71.8 | 41.2 | 57.8 | 56.1 | 71.0 | 59.6 |
| | | PPL-1% | 71.8 | 40.9 | 57.8 | 57.3 | 70.0 | 59.6 |
| | | PPL-5% | 71.8 | 41.3 | 57.9 | 55.9 | 67.0 | 58.8 |
| | | PPL-10% | 71.6 | 41.3 | 58.1 | 55.5 | 70.0 | 59.3 |

Table 5: Evaluation scores (%) on *reading comprehension* benchmarks.

| | | | BoolQ | LAMBADA | TyDiQA | TyDiQA | Avg. |
|---|---|---|---|---|---|---|---|
| | | | acc | acc | F1 | acc | |
| 0.1B | ReLU | dense | 60.8 | 30.1 | 17.9 | 4.1 | 28.2 |
| | | PPL-1% | 60.6 | 28.5 | 19.9 | 4.5 | 28.4 |
| | | PPL-5% | 60.6 | 25.6 | 17.9 | 3.4 | 26.9 |
| | | PPL-10% | 60.1 | 24.6 | 16.4 | 3.9 | 26.2 |
| | SiLU | dense | 56.5 | 31.4 | 18.5 | 4.5 | 27.7 |
| | | PPL-1% | 56.2 | 31.1 | 19.1 | 5.5 | 28.0 |
| | | PPL-5% | 53.6 | 28.9 | 18.0 | 5.5 | 26.5 |
| | | PPL-10% | 51.9 | 25.7 | 16.6 | 5.0 | 24.8 |
| 0.2B | ReLU | dense | 56.3 | 38.4 | 38.0 | 30.0 | 40.7 |
| | | PPL-1% | 56.2 | 35.8 | 36.8 | 30.0 | 39.7 |
| | | PPL-5% | 56.4 | 33.0 | 36.3 | 28.6 | 38.6 |
| | | PPL-10% | 55.9 | 30.8 | 37.4 | 30.2 | 38.6 |
| | SiLU | dense | 57.5 | 38.7 | 36.3 | 28.2 | 40.2 |
| | | PPL-1% | 57.5 | 38.3 | 35.3 | 27.5 | 39.6 |
| | | PPL-5% | 55.2 | 36.0 | 31.6 | 24.3 | 36.8 |
| | | PPL-10% | 54.5 | 34.0 | 28.1 | 20.9 | 34.4 |
| 0.4B | ReLU | dense | 61.7 | 42.9 | 43.6 | 28.0 | 44.0 |
| | | PPL-1% | 61.6 | 41.3 | 42.1 | 26.6 | 42.9 |
| | | PPL-5% | 60.8 | 39.1 | 39.9 | 23.4 | 40.8 |
| | | PPL-10% | 60.2 | 37.8 | 39.2 | 22.5 | 39.9 |
| | SiLU | dense | 57.6 | 43.0 | 41.1 | 25.4 | 41.8 |
| | | PPL-1% | 56.6 | 43.1 | 40.5 | 23.4 | 40.9 |
| | | PPL-5% | 55.2 | 39.2 | 38.1 | 20.4 | 38.2 |
| | | PPL-10% | 52.7 | 35.9 | 35.0 | 17.7 | 35.3 |
| 0.8B | ReLU | dense | 62.1 | 47.3 | 42.6 | 27.3 | 44.8 |
| | | PPL-1% | 61.7 | 45.7 | 41.0 | 24.6 | 43.2 |
| | | PPL-5% | 60.9 | 43.8 | 40.0 | 24.1 | 42.2 |
| | | PPL-10% | 59.8 | 42.5 | 37.8 | 21.1 | 40.3 |
| | SiLU | dense | 63.1 | 46.9 | 41.0 | 22.1 | 43.3 |
| | | PPL-1% | 63.1 | 46.0 | 43.3 | 24.8 | 44.3 |
| | | PPL-5% | 62.5 | 44.7 | 37.5 | 18.2 | 40.7 |
| | | PPL-10% | 62.7 | 43.0 | 34.6 | 15.0 | 38.8 |
| 1.2B | ReLU | dense | 63.3 | 52.5 | 54.3 | 42.5 | 53.2 |
| | | PPL-1% | 63.4 | 52.2 | 55.0 | 42.7 | 53.3 |
| | | PPL-5% | 62.1 | 49.5 | 56.3 | 45.2 | 53.3 |
| | | PPL-10% | 62.6 | 47.7 | 56.8 | 44.5 | 52.9 |
| | SiLU | dense | 63.2 | 53.4 | 55.2 | 47.3 | 54.8 |
| | | PPL-1% | 63.7 | 54.2 | 56.1 | 47.5 | 55.4 |
| | | PPL-5% | 62.2 | 51.2 | 53.1 | 43.9 | 52.6 |
| | | PPL-10% | 60.2 | 47.5 | 53.1 | 43.4 | 51.1 |

Table 6: Evaluation scores (%) on other more complex benchmarks.

| | | | AGIEval | HumanEval | MBPP | GSM8K | MMLU | BBH | Avg. |
|---|---|---|---|---|---|---|---|---|---|
| | | | acc | pass@1 | pass@1 | acc | acc | acc | |
| 0.1B | ReLU | dense | 23.4 | 0.6 | 0.3 | 1.8 | 26.3 | 29.3 | 13.6 |
| | | PPL-1% | 23.3 | 0.6 | 0.3 | 1.7 | 26.5 | 29.5 | 13.7 |
| | | PPL-5% | 23.5 | 0.6 | 0.1 | 1.9 | 26.3 | 28.7 | 13.5 |
| | | PPL-10% | 23.4 | 0.0 | 0.2 | 1.4 | 26.4 | 29.7 | 13.5 |
| | SiLU | dense | 23.6 | 0.6 | 0.8 | 1.6 | 26.1 | 29.2 | 13.7 |
| | | PPL-1% | 23.5 | 0.6 | 0.4 | 2.1 | 25.6 | 28.5 | 13.4 |
| | | PPL-5% | 23.6 | 0.6 | 0.3 | 1.4 | 25.8 | 30.6 | 13.7 |
| | | PPL-10% | 23.0 | 1.2 | 0.4 | 1.4 | 25.8 | 29.0 | 13.5 |
| 0.2B | ReLU | dense | 23.2 | 2.4 | 1.5 | 1.6 | 27.2 | 28.8 | 14.1 |
| | | PPL-1% | 22.8 | 2.4 | 1.2 | 2.1 | 26.9 | 30.3 | 14.3 |
| | | PPL-5% | 22.7 | 2.4 | 1.0 | 1.6 | 27.1 | 29.7 | 14.1 |
| | | PPL-10% | 23.0 | 2.4 | 1.2 | 2.1 | 26.4 | 30.1 | 14.2 |
| | SiLU | dense | 24.2 | 4.3 | 1.0 | 2.2 | 25.7 | 29.6 | 14.5 |
| | | PPL-1% | 24.2 | 4.3 | 1.8 | 2.0 | 25.2 | 29.1 | 14.4 |
| | | PPL-5% | 23.9 | 5.5 | 1.6 | 1.4 | 25.0 | 29.0 | 14.4 |
| | | PPL-10% | 23.2 | 3.0 | 0.5 | 2.4 | 24.2 | 28.4 | 13.6 |
| 0.4B | ReLU | dense | 24.6 | 6.7 | 2.3 | 2.1 | 26.1 | 30.3 | 15.3 |
| | | PPL-1% | 24.3 | 7.9 | 3.1 | 1.9 | 26.2 | 30.1 | 15.6 |
| | | PPL-5% | 24.6 | 7.9 | 2.9 | 2.2 | 26.6 | 30.2 | 15.7 |
| | | PPL-10% | 25.0 | 7.3 | 2.7 | 2.4 | 26.5 | 29.8 | 15.6 |
| | SiLU | dense | 24.4 | 5.5 | 3.2 | 2.6 | 24.9 | 30.6 | 15.2 |
| | | PPL-1% | 24.6 | 5.5 | 3.7 | 3.3 | 25.8 | 29.4 | 15.4 |
| | | PPL-5% | 24.5 | 6.1 | 2.9 | 3.8 | 25.3 | 29.6 | 15.4 |
| | | PPL-10% | 24.2 | 4.9 | 2.3 | 2.7 | 24.6 | 30.1 | 14.8 |
| 0.8B | ReLU | dense | 25.4 | 9.2 | 5.3 | 4.2 | 26.3 | 30.1 | 16.7 |
| | | PPL-1% | 25.7 | 9.2 | 5.8 | 4.5 | 26.3 | 30.0 | 16.9 |
| | | PPL-5% | 25.3 | 8.5 | 5.4 | 4.5 | 26.5 | 29.8 | 16.7 |
| | | PPL-10% | 25.8 | 8.5 | 5.0 | 4.0 | 26.4 | 29.2 | 16.5 |
| | SiLU | dense | 25.4 | 9.2 | 4.7 | 4.1 | 24.7 | 28.9 | 16.1 |
| | | PPL-1% | 25.1 | 7.9 | 4.6 | 4.0 | 24.8 | 29.7 | 16.0 |
| | | PPL-5% | 25.1 | 7.3 | 3.8 | 3.6 | 24.5 | 29.4 | 15.6 |
| | | PPL-10% | 24.8 | 7.3 | 3.9 | 3.0 | 24.2 | 28.8 | 15.3 |
| 1.2B | ReLU | dense | 26.6 | 7.3 | 6.2 | 6.4 | 33.4 | 29.9 | 18.3 |
| | | PPL-1% | 26.5 | 9.8 | 7.8 | 7.7 | 33.9 | 30.3 | 19.3 |
| | | PPL-5% | 25.8 | 7.9 | 7.4 | 6.3 | 34.3 | 30.2 | 18.6 |
| | | PPL-10% | 25.9 | 7.3 | 6.6 | 5.9 | 34.0 | 30.6 | 18.4 |
| | SiLU | dense | 26.2 | 9.8 | 9.0 | 5.2 | 32.6 | 30.9 | 18.9 |
| | | PPL-1% | 27.0 | 11.0 | 8.9 | 5.8 | 32.2 | 30.4 | 19.2 |
| | | PPL-5% | 25.7 | 7.9 | 8.5 | 5.1 | 31.0 | 30.0 | 18.0 |
| | | PPL-10% | 25.6 | 9.2 | 6.9 | 4.0 | 30.7 | 30.1 | 17.8 |

