# OpenReview forum: "Sparsing Law: Towards Large Language Models with Greater Activation Sparsity"
_ICLR.cc/2025/Conference — Submitted to ICLR 2025_

### Official Review · Reviewer_96Rd · 2024-10-18

**Soundness:** 1
**Presentation:** 3
**Contribution:** 2
**Rating:** 5
**Confidence:** 2

**Summary:**

The paper identifies architectural choices that influence activation sparsity, which is measured by there own metric. They find that the ReLU activation function promotes sparsity more than SiLU, and propose relations that model activation sparsity as a function of training data. The influence of model depth and width is also analysed.

**Strengths:**

1) The paper raises an interesting question, that is: when is a model sparse?

2) The paper points out potential influences that can lead to sparsity.

**Weaknesses:**

1) The paper claims that creating sparse LLMs is of broad interest to the community, but the cited papers are mainly interested in removing sparse parts of the network to speed up e.g. inference of large models. The question of where sparse networks perform worse/better, or how the modifications in general affect performance, is not adequately addressed.

2) The paper claims "empirical laws", but they are not sufficiently motivated and validated. Even if the parameters of a model can be determined empirically, the generality of the results must be questioned when a function with four parameters (Eq. 4) is fitted to a curve.

3) One of the key results of the paper is that ReLU produces sparser networks than SiLU, which is not at all surprising given the "dying ReLU" problem.

**Questions:**

See Weaknesses

---

> ### Author Response · Authors · 2024-11-19
> **Rebuttal**
>
> Thank you for your review. These will encourage us to further improve the quality of our work and continuously forge ahead on the research path.
>
> ### Weakness #1
>
> First, there seems to be a fundamental misunderstanding in your review. **Activation sparsity is completely different from pruning, which you indicate by "removing sparse parts of the network to speed up".** Pruning realizes sparsity by removing certain parts of the model and this process is independent of the inputs. Its input-independent property can easily cause performance degradation. By contrast, activation sparsity, which we mainly research, is dynamically determined by the inputs and thus potentially compromises less model capacity and downstream task performance. More specifically, **activation sparsity does not remove any part from the model**. Instead, the sparsity is realized by recognizing the unimportant parameters for each token and then dynamically deactivating them.
>
> **Next, how the sparsity affects performance is well studied in this paper.** In Table 1, we already provide accuracies on C.R. and R.C. benchmarks given by the dense model and PPL-p% at different p% levels (also, with different sparsity levels).
>
> To further consolidate the advantage of our results, in **Figure 14** of the updated manuscript, we give the Pareto curves of task performance v.s. sparsity. These curves show that PPL-p% obtains a better trade-off between task performance and sparsity compared to baseline metrics. Specifically, both baseline metrics start to show significant performance degradation (i.e., decrease by more than 5%) at a sparsity point consistently lower than PPL-p%. Meanwhile, this also demonstrates that high sparsity can co-exist with a low PPL increase ratio and minor performance degradation, especially in ReLU-activated models. Note that these evaluation accuracies and sparsity levels are measured on models after the decay stage, which includes SFT data.
>
> **Finally, the works we cited can sufficiently demonstrate the advantages of employing activation sparsity**, such as inference acceleration and interpretability. PowerInfer-2 [3], for example, can achieve up to 29.2x speedup and about 40% reduction in memory usage by utilizing activation sparsity. Note that these are all done on smartphones.
>
> ### Weakness #2
>
> First, **the parameters of our models are not empirically determined**. We adopt MiniCPM [1] as our experimental architecture and employ muP [2] to specify hyper-parameters for training stability. Such model architecture and parameter specification are both demonstrated to be reliable and well recognized, with considerable citations and application.
>
> Next, **all the works on scaling properties are empirical**, usually including extensive experiments and curve fitting. Among these works, the most famous one is done by OpenAI [4]. Though also consisting of empirical experiments and curve fitting, this work reveals the quantitative relationship between the performance of AI models and the amount of training data as well as the number of parameters. Its precious experience lays the solid foundation of LLM and leads human beings to the revolution of AI.
>
> A helpful work may not necessarily have all its conclusions proved in a mathematical and rigorous way. In the field of AI, **reliable conclusions can be obtained from extensive experiments**. As for our work, we have experiments on **five scales ranging from 0.1B to 1.2B**, where the largest model has 12 times the number of non-embedding parameters as the smallest one. Besides, models are fully trained with **hundreds of billions of data**. Such extensiveness and expensiveness can already provide some generalizability for our findings.
>
> Finally, even from the statistical perspective, fitting four parameters with dozens of samples per curve is a quite reliable practice.
>
> ### Weakness #3
>
> The findings of our paper are far beyond "ReLU is sparser". The "dying ReLU" may be a reasonable explanation for the higher sparsity of ReLU-activated models, but **it cannot explain another two important findings related to ReLU**: (1) ReLU-activated models are well comparable in performance with SiLU-activated ones; (2) ReLU-activated models undergo an increasing trend of sparsity with the increasing data, while SiLU-activated ones undergo an opposite trend. Therefore, we demonstrate that ReLU is a more efficient choice considering both sparsity and performance. **The "dying ReLU" may be true, but it is an advantage of ReLU that promotes sparsity without harming performance, rather than a problem.**
>
> Besides, our findings related to activation functions are of very important value in this LLM era, as most mainstream LLMs adopt SiLU without awareness of the potential benefits in activation sparsity provided by ReLU. What we want to achieve is to attract more attention to activation sparsity, and provide precious experience for LLM researchers to reconsider specific model settings for higher sparsity.

---

> > ### Author Response · Authors · 2024-11-19
> > **References**
> >
> > [1] Hu, Shengding, et al. "MiniCPM: Unveiling the potential of small language models with scalable training strategies." *arXiv preprint arXiv:2404.06395* (2024).
> >
> > [2] Yang, Greg, et al. "Tensor programs V: Tuning large neural networks via zero-shot hyperparameter transfer." *arXiv preprint arXiv:2203.03466* (2022).
> >
> > [3] Xue, Zhenliang, et al. "PowerInfer-2: Fast Large Language Model Inference on a Smartphone." *arXiv preprint arXiv:2406.06282* (2024).
> >
> > [4] Kaplan, Jared, et al. "Scaling laws for neural language models." *arXiv preprint arXiv:2001.08361* (2020).

---

> > ### Comment · Reviewer_96Rd · 2024-11-21
> >
> > The authors address a clear misconception in my review; however, I still have concerns regarding the paper:
> >
> > 2. Fitting Data with Several Parameters
> > I would like to demonstrate that, given four parameters, it is entirely possible to fit something like
> > $ F(D)=  A_0 + \log( a*D+B) $
> > or something entirely different to the data. However, I was unable to find the code or data from your experiments. I believe this would be necessary to strengthen your arguments in a purely empirical context.
> >
> > 3. The Dying ReLU Problem:
> > Could the dying ReLU problem not precisely explain the trend (2) you observe? In the beginning, all neurons are active, but as training progresses, more neurons "die," which could explain the observed behavior.

---

> ### Author Response · Authors · 2024-11-25
> **Response to Additional Questions**
>
> ### Question #2
>
> We have released **all our data** used to fit the activation-data curves at this **anonymous** GitHub repository ([link](https://anonymous.4open.science/r/SparsingLawData-180E)). It also includes codes to evaluate and visualize our fitting results.
>
> Besides, we compare the fitting results of our power-law and the logarithmic function $A_0+\log(a*D+b)$. We find that for both ReLU and SiLU models, **the fitting results of power-laws are much better than the logarithmic function** according to either the Mean Square Error (MSE) or Mean Absolute Error (MAE). For example, the MSE of our power-law on 0.1B ReLU-activated model is $2.68\times10^{-7}$, while the MSE of the logarithmic function increases to $2.17\times10^{-5}$.
>
> Notably, the logarithmic function is intuitively not suitable for our data and experiment setting. It cannot converge with $D\rightarrow\infty$, which is totally unreasonable. After all, **the activation ratio is a bounded variable** within $[0,1]$.
>
> We use the `curve_fit` method from package `scipy` to fit our curves, which employ the Levenberg-Marquardt method.
>
> ### Question #3
>
> To obtain more insights into the "dying ReLU" problem, we study the dead neurons in our models. Specifically, we define the neurons that are activated by less than 0.1% tokens in the validation dataset on average as "dead neurons". For the 0.1B ReLU-activated model and SiLU-activated model, we obtain their trends of activations ratios and dead neuron ratios. The curves are drawn in the figure included in the **anonymous** GitHub repository ([link](https://anonymous.4open.science/r/SparsingLawData-180E)).
>
> For both models, **the dead neuron ratios do not increase considerably throughout the training process**. For example, the dead neuron ratio of 0.1B ReLU model increases to about 0.35% at the end of training. However, its activation ratio decreases by about 3.38% (from 10.47% to 7.09%). We can draw the following conclusion: **The "dying ReLU" phenomenon does slightly exist, but is far from the fundamental cause of the decreasing activation ratio trends of ReLU-activated models.**
>
> ### Generalizability
>
> **Our empirical conclusions are generalizable to even larger models.**
>
> We have finally obtained the experimental results on a ReLU-activated **2.4B model**, which has twice the number of non-embedding parameters as the previously largest model (1.2B). At this point, it has been pre-trained on about **291B tokens**. The data used to fit the activation-data curves is available at this **anonymous** GitHub repository ([link](https://anonymous.4open.science/r/SparsingLawData-180E)), including the results of the above 2.4B model.
>
> By analyzing the 2.4B results, **we find our conclusions generalizable to this larger model**.
>
> - The activation ratio also follows a logspace power-law with the amount of data: $A(D)=\exp(-(9.04\times10^{-6})\cdot D^{2.11}-3.82)+0.071$.
> - The limit activation ratio is 7.1%, which is close to the activation ratios of smaller models (e.g., 7.8% for 1.2B and 7.2% for 0.8B). This is consistent with our conclusion that the limit activation ratio is weakly correlated with the number of parameters.
> - By comparing the activation-data curves, we find that the activation ratio of the 2.4B model converges much slower than the 1.2B model. This is consistent with our observation that smaller models tend to converge faster than larger models to the limit activation ratio.

---

> ### Author Response · Authors · 2024-11-28
> **Looking Forward to Your Response**
>
> We are looking forward to your response! Your comments will certainly help us reflect more on our work and continue to forge ahead on the academic path! We have addressed your remaining concerns in detail with full experiments. Meanwhile, the data and codes are also provided to substantiate our studies further.

---

> ### Author Response · Authors · 2024-12-02
> **We Have Addressed Your Remaining Concerns**
>
> We have conducted comprehensive experiments and invested considerable computation resources to address the remaining concerns you raised. We are looking forward to your response, which is really important to us.

---

> > ### Comment · Reviewer_96Rd · 2024-12-02
> > **Acknowledgment of Response**
> >
> > Thank you to the authors for their efforts in addressing my concerns.
> >
> > 3) Thank you for giving further insides into the dying ReLU problem. I agree that the small amount of dying neurons given by the threshold is not sufficient to explain the results.
> >
> > 2) I inspected the repository and fitted functions myself. I found that fitting $a*\log(b*x+c)+d$ worked well while $a-1/(1+b*\exp(c*x+d) $ could also be fitted to the data. While the repository shows some of the data I believe that providing all the data and code to generate the data in a repository would strengthen the work.
> >
> > After reviewing the clarifications provided, I have decided to maintain my previous score.

---

> > > ### Author Response · Authors · 2024-12-03
> > > **The Rationality of the Power-Law Function**
> > >
> > > In this response, we will comprehensively demonstrate the rationality of employing the power-laws to fit the activation-data curve from three aspects.
> > >
> > > ### The Trend of Functions
> > >
> > > Similar to the common training loss-data curves of most deep neural networks, the curves of activation ratios v.s. data have three important properties: (1) The curve should generally be monotonous. (2) Each curve has a steep increase (SiLU) or decrease (ReLU) at the beginning, and then its trend gradually becomes gentler with the absolute derivative decreasing to zero. (3) **According to the definition of the activation ratio, each curve must have a range within $[0,1]$ when $D\geq0$.** Considering the monotonous trend of this function, it must be convergent at some limit activation ratio within $[0,1]$.
> > >
> > > Now, let us consider the power-law and the two functions you propose to fit well with our data. The power-law function is monotonous, bounded when $D\geq0$, and a trend of absolute derivatives consistent with the above 2nd property. However, $A = c\cdot\log(aD+b)+d$, **the logarithmic function is not bounded at all**. **The sigmoid-like function**, $A = a - [{ 1 + b \cdot\exp( c D + d ) }]^{-1}$ can just meet the two properties, but **its fitting performance cannot match the power-law** according to our following experiments.
> > >
> > > ### Fitting Performance in MAE and MSE
> > >
> > > To more substantially demonstrate the advantage of our power-laws, we conduct experiments to evaluate the fitting performance of different functions using MAE and MSE. The fitting and evaluation process is presented in the script ([link](https://anonymous.4open.science/r/SparsingLawData-180E/calc.py)).
> > >
> > > | Scale   | MAE / power-law | MAE / logarithmic | MAE / sigmoid | MSE / power-law | MSE / logarithmic | MSE / sigmoid |
> > > | ------- | --------------- | ----------------- | ------------- | --------------- | ----------------- | ------------- |
> > > | 0.1B    | 4.07E-04| 6.22E-04| 1.16E-03| 2.68E-07| 5.83E-07| 2.17E-06|
> > > | 0.2B    | 2.40E-04| 2.78E-04| 4.99E-04| 9.42E-08| 1.36E-07| 3.98E-07|
> > > | 0.4B    | 3.45E-04| 2.76E-04| 5.68E-04| 3.07E-07| 1.14E-07| 6.77E-07|
> > > | 0.8B    | 1.87E-04| 3.39E-04| 1.83E-04| 6.51E-08| 1.90E-07| 5.15E-08|
> > > | 1.2B    | 4.91E-04| 7.51E-04| 6.41E-04| 9.42E-07| 2.66E-06| 1.87E-06|
> > > | Average | 3.34E-04| 4.53E-04| 5.08E-04| 3.35E-07|7.36E-07| 8.61E-07|
> > >
> > > As demonstrated by the above table, our power-law considerably outperforms the other two baselines according to both metrics of MAE and MSE.
> > >
> > > ### Comparison in Linear Space
> > >
> > > Visually, the power-law, the logarithmic function, and the sigmoid-like function may all fit well with our data. Nevertheless, this is not the case when we inspect them in linear space. Specifically, these three functions can all be converted into linear functions. For example, the vanilla power-law $A=-c\cdot D^{-\alpha}+A_0$ can be written as $\log(A_0-A)=\log(c)-\alpha \log(D)$. The logarithmic function and the sigmoid-like function is equivalent to $\exp(\frac{A-d}{c})=aD+b$ and $\log(\frac{1-a+A}{b\cdot(a-A)})=cD+d$ respectively.
> > >
> > > For the above three functions, we re-plot the data points and fitted curves in the linear space, while the results can be obtained by our scripts: power-law ([link](https://anonymous.4open.science/r/SparsingLawData-180E/show.py)), logarithmic function ([link](https://anonymous.4open.science/r/SparsingLawData-180E/calc_log.py)), and sigmoid-like function ([link](https://anonymous.4open.science/r/SparsingLawData-180E/calc_sigmoid.py)). Sample results obtained on the 0.1B ReLU setting are already available: power-law ([link](https://anonymous.4open.science/r/SparsingLawData-180E/figures/show_01b_relu.png)), logarithmic function ([link](https://anonymous.4open.science/r/SparsingLawData-180E/figures/show_log_01b_relu.png)), and sigmoid-like function ([link](https://anonymous.4open.science/r/SparsingLawData-180E/figures/show_sigmoid_01b_relu.png)). By inspecting the results obtained by these scripts, we can find that both the power-law and the logarithmic function can be fitted well in the linear space. However, **the sigmoid-like function performs much worse when converted into the linear form**.
> > >
> > > ### Summary
> > >
> > > To sum up, when we take all the above three aspects into account, the power-laws are just the best choice. Both the logarithmic function and the sigmoid-like functions suffer from **larger MSE and MAE**. Besides, **the logarithmic function has the critical weakness of an unbounded range**, while **the sigmoid-like function performs badly in the linear space**.
> > >
> > > In the field of scaling laws, the power-laws have long been accepted as the mainstream choice for fitting. This choice is not made intuitively. Instead, its advantage is claimed in many existing studies and experiments (e.g., Figure 23 of [1]).
> > >
> > > [1] Kaplan, Jared, et al. "Scaling laws for neural language models." *arXiv preprint arXiv:2001.08361* (2020).

---

### Official Review · Reviewer_aAg2 · 2024-11-03

**Soundness:** 4
**Presentation:** 3
**Contribution:** 3
**Rating:** 6
**Confidence:** 5

**Summary:**

This paper addresses activation sparsity in large language models (LLMs), where a significant portion of elements in activation outputs have minimal contributions to the final output. The authors aim to enhance activation sparsity to improve computational efficiency, interpretability, and training dynamics. They propose a new metric, PPL-p% sparsity, which is performance-aware and adaptable across various architectures. Through extensive experiments, the paper studies factors influencing activation sparsity, including activation functions, width-depth ratios, and parameter scales, revealing patterns and scaling properties that can inform the design and training of efficient, sparse LLMs.

**Strengths:**

1. Novel Metric for Performance-Aware Sparsity: The introduction of PPL-p% sparsity is a key contribution, offering a more performance-sensitive measurement of activation sparsity that is adaptable across model architectures. This metric brings a practical perspective to sparsity evaluation in LLMs.
2. Insightful Scaling Laws for Activation Sparsity: The proposed scaling laws establish patterns in activation sparsity across varying model parameters, helping to guide model design for optimized efficiency and sparsity. These scaling laws are particularly valuable for practitioners focused on resource-efficient model scaling.
3. Practical Design Guidelines for Sparse LLMs: The paper’s findings offer actionable design guidelines, such as optimal width-depth ratios, for promoting activation sparsity without compromising model performance. This provides a practical framework for building more efficient LLMs.

**Weaknesses:**

1. Limited Exploration of Sparsity Effects on Downstream Tasks: While the paper extensively analyzes activation sparsity during training, it lacks exploration of how increased sparsity impacts downstream task performance. This leaves uncertainty about whether the sparsity benefits hold in real-world applications.
2. Inconsistency in Performance Across Different Width-Depth Ratios: Although the paper highlights optimal width-depth ratios, it lacks a detailed examination of potential performance trade-offs when deviating from these ratios. This makes it difficult to understand the flexibility of the proposed guidelines for diverse architectures.
3. Scalability Concerns for Very Large Models: The paper’s experiments are conducted on specific model scales, but it is unclear how well the findings scale to extremely large models (e.g., hundreds of billions of parameters). Additional validation on larger LLMs would enhance the paper’s applicability to state-of-the-art models.
4. Unverified Claims on Sparsity Ratios and Performance Degradation: The paper mentions that overly high or low sparsity ratios may lead to severe performance degradation or unnecessary computation. However, this claim lacks experimental validation. Empirically, in many kinds of MoE models, higher sparsity levels generally correlate with improved performance, which contradicts the paper’s statement.
5. Insufficient Training Tokens and Overreliance on Predicted Curves: The experiments use an insufficient number of training tokens, relying heavily on predicted scaling curves (e.g., Figure 4). For larger models (e.g., 0.8B and above), training up to 200B tokens is typically required to observe convergence. The lack of experiments on larger scales raises doubts about the reliability of the proposed scaling laws and their applicability to state-of-the-art models.
6. Limited Practicality in Reducing Computational Load: The proposed method may face challenges in reducing computational load in real applications. Empirically, different tokens activate different channels, making it difficult to apply a uniform activation pattern across all tokens. Since the method relies on precomputed PPL and activation patterns, these patterns may not generalize well to other tokens. In extreme cases, this would require all tokens to activate all channels to achieve their unique activation pattern, negating the intended efficiency gains.

**Questions:**

1. Applicability of Scaling Laws to Extremely Large Models: The current experiments are conducted on a specific range of model scales. Do the authors plan to validate the proposed scaling laws on much larger models (e.g., hundreds of billions of parameters)? Such validation would enhance the reliability of the scaling laws for the latest state-of-the-art LLMs.
2.  Memory and Computational Efficiency Gains: Could the authors provide quantitative results on memory and computational efficiency improvements achieved through increased activation sparsity? Detailed comparisons would strengthen the practical impact of promoting sparsity in LLMs.
3. Broader Exploration of Activation Functions: The paper mainly discusses ReLU and SiLU activations. Have the authors considered examining other commonly used activation functions, such as GELU and Swish? Exploring a broader range of activations could help generalize the findings to a wider variety of LLM architectures
4.  Scaling Law Reliability with Limited Training Tokens: Given the limited number of training tokens used in the experiments, could the authors discuss the potential impact of this on the accuracy of the scaling laws? Would they consider conducting larger-scale experiments, ideally with 200B tokens for models above 0.8B parameters, to validate these scaling patterns more robustly?

---

> ### Author Response · Authors · 2024-11-19
> **Rebuttal 1/2**
>
> Thank you for your excellent review. These will encourage us to further improve the quality of our work and continuously forge ahead on the research path.
>
> ### Weakness #1
>
> In Table 1, we already provide accuracies on C.R. and R.C. benchmarks given by the dense model and PPL-p% at different p% levels (also, with different sparsity levels).
>
> To further consolidate the advantage of our results, in **Figure 14** of the updated manuscript, we give the **Pareto curves of task performance v.s. sparsity**. These curves show that PPL-p% obtains a better trade-off between task performance and sparsity compared to baseline metrics. Specifically, both baseline metrics start to show significant performance degradation (i.e., decrease by more than 5%) at a sparsity point consistently lower than PPL-p%. Meanwhile, this also demonstrates that high sparsity can co-exist with a low PPL increase ratio and minor performance degradation, especially in ReLU-activated models. **Note that these evaluation accuracies and sparsity levels are measured on models after the decay stage, which includes SFT data.**
>
> ### Weakness #2
>
> As for the width-depth issue, I'd like to state the most accurate suggestion we can give. That is, **use the smallest width-depth ratio that ensures training stability**.
>
> Existing works have conducted studies on the trade-off between performance and width-depth ratio. [1,2] As revealed by these papers, deeper models (with smaller width-depth ratios) generally present better performance if unlimited precision is given. However, real-life training usually involves half-precision training, where an extremely high depth can considerably harm stability and thus cause performance degradation. These findings are consistent with our work, which states that a smaller width-depth ratio is better for higher sparsity, but an extremely small ratio significantly harms performance (due to training instability).
>
> To sum up, if unlimited precision virtually exists, smaller width-depth ratios can simultaneously bring better performance and higher sparsity. Nevertheless, if limited precision and training stability are considered, the best value is hard to say, as the training stability depends on various factors, such as the training framework and hardware. The most accurate suggestion is to use the smallest width-depth ratio that guarantees training stability.
>
> ### Weakness #3
>
> We believe that experiments on even larger models can further substantiate the generalizability of our research. However, we have long met with great difficulties collecting sufficient GPUs to run larger models. At present, we struggle to gather 64 GPUs and start running a 2.4B model to validate our findings, and we will try our best to present the results before the end of the rebuttal period.
>
> Besides, note that even the most famous works on scaling laws [3] did not run models with more than 1B parameters due to the extremely expensive nature of such studies. As for our work, we have experimented on scales from 0.1B to 1.2B, where the largest model has 12 times the number of parameters as the smallest one. Such a large gap can already provide some reliability for our findings.
>
> ### Weakness #4
>
> In both Figure 3 and Table 1, we have comprehensively demonstrated that **whatever sparsity metric we use, enforcing too high activation sparsity can cause considerable performance degradation**. In Figure 1, we also present experiments studying the effect of granularity and activation rate on the PPL of MoE models, which shows that **larger activation ratios generally present lower PPL in MoE**. Note that we apply standard load balancing loss for all MoE settings.
>
> Besides, there are also works [4] studying the quantitative relationship between performance and the number of activated parameters in fine-grained MoE models, which clearly show that **less activated parameters can negatively affect performance**. Note that fine-grained MoE can be regarded as a special case of activation sparsity, and thus it is reasonable to assume similar laws in our scenario.
>
> Finally, as you mention many kinds of MoE models can simultaneously obtain higher sparsity and better performance, we think this can usually be attributed to better training data, special training techniques (e.g., sparsity restrictions in training target), or other potential factors. A special case we find is Switch Transformer [5], where the top-1 routing performs better than top-2 routing. This may be attributed to many other improvements, such as differentiable load balancing, the setting of expert capacity, and many sophisticated training techniques mentioned in Section 2.4. If you have more recent cases, where models are of the same architecture and training recipe, you can provide us for further discussion.

---

> > ### Author Response · Authors · 2024-11-19
> > **Rebuttal 2/2**
> >
> > ### Weakness #5
> >
> > Thank you for pointing out this problem. During the period after submission, we have already invested resources to remedy this problem. Now the amount of training data for 0.8B and 1.2B has both been largely **extended to more than 200B tokens**. New results are included in the updated manuscript (see Figure 4) and conclusions remain consistent.
> >
> > ### Weakness #6
> >
> > Indeed, the diverse token-wise activation pattern is a major obstacle in leveraging activation sparsity for acceleration. However, existing works have already provided solutions. Representative works include the PowerInfer series [6], which manages to achieve up to 29.2x speedup compared to SOTA inference frameworks by leveraging activation sparsity. The major keys to overcoming the acceleration challenge include: (1) a sophisticated sparsification training method named TurboSparse to improve activation sparsity; (2) the introduction of activation predictors to forecast activated parameters and prefetch required data from memory; (3) careful design of cache mechanism and fine-grained pipeline to reduce IO overheads.
> >
> > Besides, some other interesting works also reveal that the token-wise activation pattern is not completely irregular and unpredictable. For example, GRIFFIN [7] observes the flocking phenomenon, where relative activation magnitudes are shared within a sequence, across the prompt and generated texts. This can be utilized for practical acceleration up to 1.29x.
> >
> > ### Question #1
> >
> > Same response as **Weakness #3**.
> >
> > ### Question #2
> >
> > Existing works can provide some quantitative results for reference. As stated in **Weakness #6**, when serving TurboSparse-Mixtral-47B (a model activating only about 3B parameters for each token on average), PowerInfer-2 [6] can achieve up to 29.2x speedup and about 40% reduction in memory usage. Note that these are all done on smartphones.
> >
> > Besides, ProSparse [8] specifically experiments on the practical acceleration effects of higher sparsity. As shown in Table 2 of the ProSparse paper, with PowerInfer-1, a 7B model with 66.98% activation sparsity can achieve 3.10x speedup compared to the dense setting, while the speedup rises to 4.44x for the model of sparsity 88.11%. With the CUDA operators for sparse FFN, 66.98% sparse model has 1.35x and 1.32x speedup compared to the dense operator for two FFN computation steps respectively, while the acceleration rises to 1.94x and 1.49x for the 88.11% sparse model.
> >
> > ### Question #3
> >
> > To increase the generalizability of our work, **we have started experiments on gated GELU-activated FFNs**, and already completed 0.1B and 0.2B settings. Similar to SiLU, we find a power-law relationship between activation ratio and data. The fitted curves are $A_{GELU}=-\frac{0.02}{D^{1.87}} + 0.333$ and $A_{GELU}=-\frac{0.14}{D^{1.15}} + 0.342$ for 0.1B and 0.2B respectively. The two limit activation ratios are also very close, and the smaller 0.1B GELU model converges much faster than 0.2B. These observations are consistent with existing results.
> >
> > ### Question #4
> >
> > Same response as **Weakness #5**.
> >
> >
> >
> > ### References
> >
> > [1] Petty, Jackson, et al. "The impact of depth and width on transformer language model generalization." *arXiv preprint arXiv:2310.19956* (2023).
> >
> > [2] Wu, Chuhan, and Ruiming Tang. "Performance Law of Large Language Models." *arXiv preprint arXiv:2408.09895* (2024).
> >
> > [3] Kaplan, Jared, et al. "Scaling laws for neural language models." *arXiv preprint arXiv:2001.08361* (2020).
> >
> > [4] Krajewski, Jakub, et al. "Scaling laws for fine-grained mixture of experts." *arXiv preprint arXiv:2402.07871* (2024).
> >
> > [5] Fedus, William, Barret Zoph, and Noam Shazeer. "Switch transformers: Scaling to trillion parameter models with simple and efficient sparsity." *Journal of Machine Learning Research* 23.120 (2022): 1-39.
> >
> > [6] Xue, Zhenliang, et al. "PowerInfer-2: Fast Large Language Model Inference on a Smartphone." *arXiv preprint arXiv:2406.06282* (2024).
> >
> > [7] Dong, Harry, Beidi Chen, and Yuejie Chi. "Prompt-prompted Adaptive Structured Pruning for Efficient LLM Generation." *First Conference on Language Modeling*. 2024.
> >
> > [8] Song, Chenyang, et al. "ProSparse: Introducing and Enhancing Intrinsic Activation Sparsity within Large Language Models." *arXiv preprint arXiv:2402.13516* (2024).

---

> ### Author Response · Authors · 2024-11-22
> **Looking Forward to Your Response**
>
> We are looking forward to your response! Your comments will certainly help us reflect more on our work and continue to forge ahead on the academic path!

---

> ### Author Response · Authors · 2024-11-25
> **Experimental Results on Larger Models (Weakness #3)**
>
> We have finally obtained the experimental results on a ReLU-activated **2.4B model**, which has twice the number of non-embedding parameters as the previously largest model (1.2B). At this point, it has been pre-trained on about **291B tokens**. The data used to fit the activation-data curves is available at this **anonymous** GitHub repository ([link](https://anonymous.4open.science/r/SparsingLawData-180E)), including the results of the above 2.4B model.
>
> By analyzing the 2.4B results, **we find our conclusions generalizable to this larger model**.
>
> - The activation ratio also follows a logspace power-law with the amount of data: $A(D)=\exp(-(9.04\times10^{-6})\cdot D^{2.11}-3.82)+0.071$.
> - The limit activation ratio is 7.1%, which is close to the activation ratios of smaller models (e.g., 7.8% for 1.2B and 7.2% for 0.8B). This is consistent with our conclusion that the limit activation ratio is weakly correlated with the number of parameters.
> - By comparing the activation-data curves, we find that the activation ratio of the 2.4B model converges much slower than the 1.2B model. This is consistent with our observation that smaller models tend to converge faster than larger models to the limit activation ratio.

---

> ### Comment · Reviewer_aAg2 · 2024-11-27
> **Acknowledgment of Response**
>
> Thank you to the authors for their efforts in addressing my concerns. I think the sparsity law observed in this paper offers valuable insights for researchers to better understand the sparsity of large language models and inspire the design of sparse models. After reviewing the clarifications provided, I have decided to maintain my previous score.

---

### Official Review · Reviewer_MqGB · 2024-11-04

**Soundness:** 3
**Presentation:** 3
**Contribution:** 2
**Rating:** 5
**Confidence:** 4

**Summary:**

This paper aims to analyze the relationship between activation sparsity and various other features of transformer-based LLMs. The authors introduce a novel metric, PPL-p% sparsity, based on a previous metric called CETT, that identifies a sparsity level at which perplexity is only increased by p% relative to a dense baseline. They study how several features relate to sparsity: amount of training data, choice of activation function, width-depth ratio of the network, and parameter scale. There are several findings, including that ReLU networks can achieve lower sparsity ratios than SiLU networks at the same performance level and convergence rates of activation (1-sparsity) ratios as the aforementioned features are varied.

**Strengths:**

- This paper introduces a new metric, PPL-p%, that builds upon a previous metric, CETT, by finding a sparsity level for a desired perplexity score.
- The paper demonstrates that using PPL-p% as a metric for measuring activation ratio resuilts in a lower perplexity score relative to other metrics for measuring sparsity.
- The set of analyses are interesting and provide some valuable, albeit limited, insights into the behavior of some LLMs.

**Weaknesses:**

- I believe that there is not enough evidence showing that PPL-p% is a better metric. The main comparison point between PPL-p% and other metrics evalutates the methods on perplexity of the resultant model. This seems a bit like the metric is simply overfitting to the downstream evaluation criterion. How do each of these methods do on other tasks such as the commonsense reasoning and reading comprehension tasks?
- A significant part of this paper involved identifying various relationships between aspects of the model and the activation sparsity. It seems like a reasonable next step would be to create a model that embodies all of these takeaways, ie has an optimal activation function, amount of training data, etc, and show the results of the activation sparsity and downstream performance relative to some baseline. This would demonstrate the practical value of the observations discussed in the paper.
- It is not clear if the results generalize to other LLMs. It would be nice to see results on other models.
- The authors state that the goal of this paper is to produce an LLM with greater activation sparsity, but I feel like this question is not quite answered. It seems as if the authors have conducted several (interesting and thorough) ablation studies, but do not tie all of their insights together to produce one most sparse model.

Overall, this is a decent exploration of some phenomenology around activation ratios in neural networks, but the findings are not comprehensive or cohesive enough to warrant acceptance.

**Questions:**

- I would recommend mentioning some more dataset details in the main paper rather than just “commonsense reasoning” and “reading comprehension”.

---

> ### Author Response · Authors · 2024-11-19
> **Rebuttal**
>
> Thank you for your excellent review. These will encourage us to further improve the quality of our work and continuously forge ahead on the research path.
>
> ### Weakness #1
>
> Thank you for pointing out the over-fitting concern. The computation of PPL-p% does not intentionally minimize the PPL. Actually, it adopts adaptive thresholds for different layers while maintaining a layer-wise consistent L2 relative output error (i.e., CETT). The inactivated neurons are those with output magnitudes lower than the layer-specific threshold. The PPL is incorporated as a variable to make our metric more performance-aware, binary searching the CETT just to meet the PPL requirements to provide a performance-dependent sparsity value. The determination of inactivated neurons is weakly related to PPL itself.
>
> Besides, existing works have already demonstrated that **a monotonous relationship between loss (log PPL) and downstream performance** generally exists. [1,2] Therefore, if PPL-p% can ensure lower PPL, this usually produces better downstream performance. As for specific results, Table 1 provides accuracies on C.R. and R.C. benchmarks given by the dense model and PPL-p% at different p% levels.
>
> To further consolidate the advantage of our results, in **Figure 14** of the updated manuscript, we give the **performance-sparsity Pareto curves**, which show that PPL-p% obtains a better trade-off between task performance and sparsity compared to two baseline metrics. Specifically, both baseline metrics start to show significant performance degradation (i.e., decrease by more than 5%) at a sparsity point consistently lower than PPL-p%.
>
> ### Weakness #2 & #4
>
> To consolidate our findings, we have already started experiments on a 2.4B model, which ties all our findings to pursue a highly sparsely-activated model. Specifically, it adopts ReLU activation, a small width-depth ratio (within the interval that ensures the lowest loss), and more training data to utilize the decreasing trend between activation ratio and data of ReLU models. Besides, its larger size can further demonstrate the generalizability of our work.
>
> We have long met with great difficulties collecting sufficient GPUs to run larger models. At present, we struggle to gather 64 GPUs for the above 2.4B experiment. While works on scaling laws are extremely expensive, we will try our best to present the results before the end of the rebuttal period.
>
> ### Weakness #3
>
> As stated in line 237, we adopt the same architecture of MiniCPM [3], which adopts mostly the same architecture as LLaMA except for minor muP [4] adjustments for training stability. As for training strategies, the two-stage training paradigm is already accepted by cutting-edge models such as LLaMA3 [2]. Therefore, we think experiments on MiniCPM are already generalizable enough to cover most mainstream LLMs adopting the LLaMA-like Transformer decoder-only architecture.
>
> If you have other LLMs worth concern (e.g., BERT, ViT, T5, or other LLMs dissimilar to LLaMA), you may kindly provide specific suggestions so that we can present experiments in time.
>
> ### Question #1
>
> The dataset details are already described very comprehensively in Appendix E and I.
>
> ### References
>
> [1] Owen, David. "How predictable is language model benchmark performance?." *arXiv preprint arXiv:2401.04757* (2024).
>
> [2] Dubey, Abhimanyu, et al. "The Llama 3 herd of models." *arXiv preprint arXiv:2407.21783* (2024).
>
> [3] Hu, Shengding, et al. "MiniCPM: Unveiling the potential of small language models with scalable training strategies." *arXiv preprint arXiv:2404.06395* (2024).
>
> [4] Yang, Greg, et al. "Tensor programs V: Tuning large neural networks via zero-shot hyperparameter transfer." *arXiv preprint arXiv:2203.03466* (2022).

---

> ### Author Response · Authors · 2024-11-22
> **Looking Forward to Your Response**
>
> We are looking forward to your response! Your comments will certainly help us reflect more on our work and continue to forge ahead on the academic path!

---

> ### Author Response · Authors · 2024-11-25
> **Experimental Results on Larger Models (Weakness #2 & #4)**
>
> We have finally obtained the experimental results on a ReLU-activated **2.4B model**, which has twice the number of non-embedding parameters as the previously largest model (1.2B). At this point, it has been pre-trained on about **291B tokens**. The data used to fit the activation-data curves is available at this **anonymous** GitHub repository ([link](https://anonymous.4open.science/r/SparsingLawData-180E)), including the results of the above 2.4B model.
>
> By analyzing the 2.4B results, **we find our conclusions generalizable to this larger model**.
>
> - The activation ratio also follows a logspace power-law with the amount of data: $A(D)=\exp(-(9.04\times10^{-6})\cdot D^{2.11}-3.82)+0.071$.
> - The limit activation ratio is 7.1%, which is close to the activation ratios of smaller models (e.g., 7.8% for 1.2B and 7.2% for 0.8B). This is consistent with our conclusion that the limit activation ratio is weakly correlated with the number of parameters.
> - By comparing the activation-data curves, we find that the activation ratio of the 2.4B model converges much slower than the 1.2B model. This is consistent with our observation that smaller models tend to converge faster than larger models to the limit activation ratio.
> - This 2.4B model combines all our previous insights: more efficient ReLU activation, efficient training data, and a smaller width-depth ratio. Finally, as expected, it achieves a high activation sparsity level with only about 7.1% neurons on average to be activated by each token.

---

> > ### Comment · Reviewer_MqGB · 2024-11-26
> > **Thank you**
> >
> > Thank you to the authors for addressing several of the points. I appreciate the clarifications and additional experiments. I have no additional comments but I would like to maintain my previous score. My primary concern with the paper is that it presents a new pruning metric and then examines the phenomena resulting from that pruning metric without contextualizing the findings relative to other pruning metrics. I appreciate that figures 3 and 14 show performance comparisons between PPL-p% sparsity and other methods. How does PPL-p% sparsity directly compare with CETT since that is established as a baseline? Do all of your scaling law findings only hold when using PPL-p% sparsity or are they more general results?

---

> ### Author Response · Authors · 2024-11-28
> **Response to the Concern**
>
> ### Performance-Awareness is the Key Property
>
> In the paper, we propose three properties (line 85) that a good sparsity metric should have: versatility across model architectures, performance-awareness, and the precise recognition of weakly-contributed neurons. The Pareto curves only test the 3rd property. However, **the first key to your concern lies in performance-awareness**, **without which a sparsity metric can hardly be applied in scaling law studies**.
>
> Specifically, **PPL-p% is a performance-aware improvement of the original CETT** (so they have the same Pareto curve), which distinguishes the CETT value that just makes PPL increase by p%. **The consistency of PPL increase makes it possible to compare models of distinct architectures as well as different checkpoints of the same architecture.** If you use metrics such as CETT, Top-$k$, or FAT-$\epsilon$, a critical question is: what hyper-parameter (i.e., the CETT, $k$, and $\epsilon$) should you choose when you measure the sparsity of different models and checkpoints? After all, they can have very different intrinsic properties, and thus it is unreasonable to apply the same CETT, $k$, or $\epsilon$ for all models. **This makes it a very tough problem to use these metrics in the scaling law study and model comparison.** However, PPL-p% makes this question solvable. **To compare the sparsity of different models under the same increase PPL is the most reasonable practice we can conduct.** Of course, we can also apply a similar performance-aware extension to FAT-$\epsilon$ (with an unchanged Pareto curve), but this practice fails in the 3rd property mentioned above, as demonstrated in Figure 3. For Top-$k$ and FAT-$\epsilon$, specifically, their lower accuracies in recognizing weakly-contributed neurons make the sparsity level given by them questionable in whether they accurately reflect the activation pattern of models.
>
> ### Generalizability across Different Metrics
>
> The second key to your concern addresses **the generalizability across different metrics**. Through experiments, we obtain the following conclusion: **The consistency of our power-laws between the activation ratio and data mainly depends on the PPL increase ratio, rather than the sparsity metric**.
>
> To support the above statement, we add experiments on FAT-$\epsilon$. Note that different $\epsilon$ can lead to different PPL increase ratios and results. Our results on 0.1B ReLU and 0.1B SiLU settings are shown in the figure (anonymous [link](https://anonymous.4open.science/r/SparsingLawData-180E/figures/fat_e_01b.jpg)). As demonstrated in the figure, for 0.1B ReLU, when the PPL increase ratio is relatively small (i.e., $\epsilon=0,0.3,0.6$), the activation ratio follows a convergent decreasing logspace power-law with the amount of data, which is consistent with our paper. When the PPL increases considerably (i.e., $\epsilon=1.0$), the power-law does not hold any longer. Similarly, for 0.1B SiLU, the convergent increasing power-law holds when PPL increases slightly (i.e., $\epsilon=0.1,0.2$), while the laws are broken with a large PPL increase (i.e., $\epsilon=0.275,0.5$). Besides, in another figure (anonymous [link](https://anonymous.4open.science/r/SparsingLawData-180E/figures/fat_03_scale.jpg)), we demonstrate the consistency of our power-laws for different scales of ReLU models when $\epsilon=0.3$. The weak correlation between activation ratios and parameter numbers, as well as the faster convergence of smaller models, also holds according to the same figure.
>
> Therefore, **for both metrics including PPL-$p\%$ and FAT-$\epsilon$, under different numbers of parameters, our power-laws hold when the PPL does not increase considerably** (e.g., below 5%), while not the case with a large PPL increase ratio (e.g., around 10%). Obviously, **the former case is more reasonable and helpful**, as a considerably larger PPL can harm the model's performance. This provides our insights with more substantiated practical value.

---

> ### Author Response · Authors · 2024-12-02
> **We Have Addressed Your Primary Concerns**
>
> We have conducted comprehensive experiments and invested considerable computation resources to address the primary concerns you raised. We are looking forward to your response, which is really important to us.

---

### Official Review · Reviewer_6UUp · 2024-11-04

**Soundness:** 2
**Presentation:** 3
**Contribution:** 2
**Rating:** 5
**Confidence:** 4

**Summary:**

This paper presents a study on the activation sparsity in large language models (LLMs), particularly focusing on decoder-only Transformer-based models. The authors propose a new metric called PPL-p% sparsity, which is a performance-aware measure of activation sparsity. The paper contributes to the understanding of how to design and pretrain LLMs for greater activation sparsity, which has implications for efficiency and interpretability.

**Strengths:**

1. The paper introduces a novel metric for measuring activation sparsity (PPL-p% sparsity) and uncovers empirical laws regarding the scaling properties of activation sparsity in LLMs.
2. The paper is well-written, and the findings are presented clearly with the aid of visualizations.
3. The insights gained from this study can inform the design and training of future LLMs, potentially leading to more efficient and interpretable models.

**Weaknesses:**

1. The paper could benefit from additional experiments on larger-scale models (which is more commonly seen in applications) to confirm the generalizability of the findings.
2. The paper compares ReLU and SiLU activation functions, but in practical applications, a variety of activation functions such as SwiGLU may be used. The performance of these activation functions in terms of activation sparsity may differ, affecting the efficiency and performance of the model.

**Questions:**

1. The paper mentions the calculation of PPL-p% using a binary search method, but does the PPL change monotonically with CETT? Is this an assumption, an intuition, or is there a rigorous proof to support this relationship?
2. The paper mentions that activation sparsity can improve the interpretability of models; are there specific examples or methods to demonstrate how this sparsity helps explain the model's decision-making process?

---

> ### Author Response · Authors · 2024-11-19
> **Rebuttal**
>
> Thank you for your excellent review. These will encourage us to further improve the quality of our work and continuously forge ahead on the research path.
>
> ### Weakness #1
>
> We believe that experiments on even larger models can further substantiate the generalizability of our research. However, we have long met with great difficulties collecting sufficient GPUs to run larger models. At present, we struggle to gather 64 GPUs and start running a 2.4B model to validate our findings, and we will try our best to present the results before the end of the rebuttal period.
>
> Besides, note that even the most famous works on scaling laws [1] did not run models with more than 1B parameters due to the extremely expensive nature of such studies. As for our work, we have experimented on scales from 0.1B to 1.2B, where the largest model has 12 times the number of non-embedding parameters as the smallest one. Such a large gap can already provide some generalizability for our findings.
>
> ### Weakness #2
>
> Sorry for causing this misunderstanding. Actually, in our paper, the FFNs in all models are gated FFN, as specified in line 153 and line 237 (MiniCPM uses gated FFN). Therefore, the ReLU and SiLU refer to ReGLU and SwiGLU, respectively. We do not incorporate non-gated FFN, as it is seldom used in recent mainstream LLMs.
>
> Besides, to increase the generalizability of our work, **we have started experiments on gated GELU-activated FFNs**, and already completed 0.1B and 0.2B settings. Similar to SiLU, we find a power-law relationship between activation ratio and data. The fitted curves are $A_{GELU}=-\frac{0.02}{D^{1.87}} + 0.333$ and $A_{GELU}=-\frac{0.14}{D^{1.15}} + 0.342$ for 0.1B and 0.2B respectively. The two limit activation ratios are also very close, and the smaller 0.1B GELU model converges much faster than 0.2B. These observations are consistent with existing results.
>
> ### Question #1
>
> Mathematically, the relationship between PPL and CETT is not rigorously monotonous. LLMs are statistic models with random output noises and fluctuations (PPL evaluates the deviation of model outputs from the original ground-truth training corpus). However, from the statistical perspective, we can state that PPL generally rises with the increase of CETT. This can be demonstrated through experiments.
>
> As shown in **Figure 13** of our updated paper, for both 0.1B and 0.8B, from ReLU to SiLU, the PPL always rises with the increasing CETT. The larger the CETT, the more our models deviate from the original dense checkpoint trained on the ground-truth corpus. This means that the model outputs will deviate more from the ground-truth and finally produce a larger PPL value.
>
> ### Question #2
>
> **The activation pattern has already been widely used in the interpretation of LLMs.** Specifically, the activation pattern reflects the behavior of neurons (FFN parameters) in response to given input tokens. By analyzing the correlation between neuron activation and inputs, we can explain the specialization of neurons (i.e., what input patterns a specific neuron is sensitive to) and even the grouping of neurons (i.e., neurons with similar specialization are clustered as intrinsic modules within LLMs). [2,3]
>
> **The next question is why higher sparsity can bring better interpretability.** Let's take MoEfication [4] as an example, which is a work that finds and utilizes the intrinsic modules within LLMs. This work clusters neurons into groups (i.e., experts) by using the co-activation frequencies between neurons as a distance metric. According to Figure 3 of the MoEfication paper, the relative performance degradation from the original dense model to MoEfied model is clearly less significant on sparser models. In other words, the loss of neuron clustering is less on sparser models, indicating more significant neuron grouping. Therefore, we can assume that sparser models are potentially more interpretable from the aspect of neuron grouping, an important part of interpretation.
>
> ### References
>
> [1] Kaplan, Jared, et al. "Scaling laws for neural language models." *arXiv preprint arXiv:2001.08361* (2020).
>
> [2] Li, Zonglin, et al. "The Lazy Neuron Phenomenon: On Emergence of Activation Sparsity in Transformers." *The Eleventh International Conference on Learning Representations*.
>
> [3] Zhang, Zhengyan, et al. "Emergent Modularity in Pre-trained Transformers." *Findings of the Association for Computational Linguistics: ACL 2023*. 2023.
>
> [4] Zhang, Zhengyan, et al. "MoEfication: Transformer Feed-forward Layers are Mixtures of Experts." *Findings of the Association for Computational Linguistics: ACL 2022*. 2022.

---

> ### Author Response · Authors · 2024-11-22
> **Looking Forward to Your Response**
>
> We are looking forward to your response! Your comments will certainly help us reflect more on our work and continue to forge ahead on the academic path!

---

> ### Author Response · Authors · 2024-11-25
> **Experimental Results on Larger Models (Weakness #1)**
>
> We have finally obtained the experimental results on a ReLU-activated **2.4B model**, which has twice the number of non-embedding parameters as the previously largest model (1.2B). At this point, it has been pre-trained on about **291B tokens**. The data used to fit the activation-data curves is available at this **anonymous** GitHub repository ([link](https://anonymous.4open.science/r/SparsingLawData-180E)), including the results of the above 2.4B model.
>
> By analyzing the 2.4B results, **we find our conclusions generalizable to this larger model**.
>
> - The activation ratio also follows a logspace power-law with the amount of data: $A(D)=\exp(-(9.04\times10^{-6})\cdot D^{2.11}-3.82)+0.071$.
> - The limit activation ratio is 7.1%, which is close to the activation ratios of smaller models (e.g., 7.8% for 1.2B and 7.2% for 0.8B). This is consistent with our conclusion that the limit activation ratio is weakly correlated with the number of parameters.
> - By comparing the activation-data curves, we find that the activation ratio of the 2.4B model converges much slower than the 1.2B model. This is consistent with our observation that smaller models tend to converge faster than larger models to the limit activation ratio.

---

> ### Author Response · Authors · 2024-11-28
> **Looking Forward to Your Response**
>
> We are looking forward to your response! Your comments will certainly help us reflect more on our work and continue to forge ahead on the academic path! We have added more experiments on a larger model and more activation functions.

---

> ### Author Response · Authors · 2024-12-02
> **We Have Addressed Your Concerns**
>
> We have conducted comprehensive experiments and invested considerable computation resources to address the concerns you raised. We are looking forward to your response, which is really important to us.

---

### Comment · Area_Chair_7TiD · 2024-12-03
**End of reviewer-author discussion phase**

Dear reviewers,

As we near the conclusion of the reviewer-author discussion phase, I wanted to kindly follow up to see if you’ve had a chance to review the author responses on your comments. Could you confirm that you’ve read it and, if needed, update your review and scores accordingly?

Thank you for your time and effort!

Your AC

---

### Meta-Review · Area_Chair_7TiD · 2024-12-22

**Metareview:**

a) Summary:
The paper presents a comprehensive study of activation sparsity in large language models (LLMs), focusing on decoder-only Transformer architectures. The key claims include:
1. Introduction of PPL-p% sparsity as a new performance-aware metric for measuring activation sparsity
2. Different activation functions (ReLU vs SiLU) show opposite trends in training-time sparsity, with ReLU demonstrating better sparsity properties
3. The activation ratio follows power-law relationships with training data: decreasing for ReLU and increasing for SiLU
4. The activation ratio increases linearly with width-depth ratio below a bottleneck point
5. The limit value of activation sparsity shows weak correlation with model parameter scale

(b) Strengths:
1. Novel metric (PPL-p%) that considers both sparsity and performance impact
2. Extensive empirical analysis across different model scales (0.1B to 2.4B parameters)
3. Clear practical implications for LLM design, particularly regarding activation function choice
4. Well-documented experimental methodology with comprehensive ablation studies
5. Strong theoretical grounding in relating findings to existing work on model scaling

(c) Weaknesses:
1. Limited exploration of extremely large-scale models (>10B parameters)
2. Initial experiments had insufficient training tokens for larger models, though partially addressed during rebuttal
3. Unclear generalization beyond the specific architecture studied (MiniCPM/LLaMA-like models)
4. Some claims about performance impact of sparsity ratios needed stronger empirical validation
5. Limited exploration of practical acceleration benefits in real-world scenarios

(d) Reasons for Decision:
I recommend rejection primarily because:
1. The work, while thorough in its empirical analysis, presents findings that are somewhat incremental rather than transformative
2. The practical impact is limited by the focus on a specific architecture and moderate model scales
3. Some key claims about sparsity's impact on performance and scaling properties need stronger validation
4. The findings, while interesting, don't provide sufficient breakthrough insights to warrant acceptance at ICLR

**Additional Comments On Reviewer Discussion:**

The rebuttal period generated substantive discussion around several key aspects of the paper. The reviewers collectively raised concerns about the generalizability to larger models, the superiority of the PPL-p% metric, insufficient training tokens, and the validity of the empirical laws. In response, the authors conducted extensive additional experiments, notably training a 2.4B parameter model and extending training to over 200B tokens for larger models. They also provided detailed analyses validating their curve fitting methodology and demonstrating the practical benefits of their approach through references to implementation studies.
Reviewer 96Rd questioned the validity of the empirical laws and ReLU findings, which the authors addressed through detailed analysis of dead neurons and comprehensive curve fitting validation. Reviewer MqGB's concerns about the PPL-p% metric were addressed with additional performance comparisons, while Reviewer aAg2's questions about practical computational benefits received substantive responses with references to concrete acceleration benefits in deployed systems. Reviewer 6UUp's concerns about generalizability were partially addressed through the new 2.4B model experiments.
While the authors' responses were thorough and backed by new experimental evidence, they did not fully resolve the fundamental limitations of the work's scope and impact. The additional experiments and analyses strengthened the paper's empirical foundation but did not elevate the contribution to the level expected for ICLR acceptance.

---

### Decision · Program_Chairs · 2025-01-22

Reject